# Active Learning of *General* Halfspaces: Label Queries *vs* Membership Queries

**Ilias Diakonikolas**[*]
University of Wisconsin-Madison
ilias@cs.wisc.edu

**Daniel M. Kane**[*]
University of California, San Diego
dakane@cs.ucsd.edu

**Mingchen Ma**[*]
University of Wisconsin-Madison
mingchen@cs.wisc.edu

## Abstract

We study the problem of learning *general* (i.e., not necessarily homogeneous) halfspaces under the Gaussian distribution on $\mathbb{R}^d$ in the presence of some form of query access. In the classical pool-based active learning model, where the algorithm is allowed to make adaptive label queries to previously sampled points, we establish a strong information-theoretic lower bound ruling out non-trivial improvements over the passive setting. Specifically, we show that any active learner requires label complexity of $\tilde{\Omega}(d/(\log(m)\epsilon))$, where $m$ is the number of unlabeled examples. Specifically, to beat the passive label complexity of $\tilde{O}(d/\epsilon)$, an active learner requires a pool of $2^{\mathrm{poly}(d)}$ unlabeled samples. On the positive side, we show that this lower bound can be circumvented with membership query access, even in the agnostic model. Specifically, we give a computationally efficient learner with query complexity of $\tilde{O}(\min\{1/p, 1/\epsilon\} + d\,\mathrm{polylog}(1/\epsilon))$ achieving error guarantee of $O(\mathrm{opt} + \epsilon)$. Here $p \in [0, 1/2]$ is the bias and $\mathrm{opt}$ is the 0-1 loss of the optimal halfspace. As a corollary, we obtain a strong separation between the active and membership query models. Taken together, our results characterize the complexity of learning general halfspaces under Gaussian marginals in these models.

## 1   Introduction

In Valiant's PAC learning model [Val84a, Val84b], the learner is given access to random labeled examples and aims to find an accurate approximation to the function that generated the labels. The standard PAC model is "passive" in the sense that the learner has no control over the selection of the training set. Here we focus on *interactive* learning between a learner and a domain expert that can potentially lead to significantly more efficient learning procedures. A standard such paradigm is (pool-based) active learning [MN⁺98], where the learner has access to a large pool of unlabeled examples $S$ and has the ability to (adaptively) select a subset of $S$ and obtain their labels. We will henceforth refer to this type of data access as *label query* access. An even stronger interactive model is that of PAC learning with *membership queries* [Ang88, Fel09]. A *membership query (MQ)* allows the learner to obtain the value of the target function on *any* desired point in the support of the marginal distribution. This model captures the ability to perform experiments or the availability of expert advice. While in active learning, the learner is allowed to query the labels of previously sampled points from $S$, in MQ learning the learner has black-box access to the target function. We refer the reader to Appendix A for formal definitions of these two learning models. Roughly speaking, when

---

[*]Equal Contribution

38th Conference on Neural Information Processing Systems (NeurIPS 2024).

the size of $S$ becomes exponentially large (so that it is a good cover of the space), the model of active learning "converges" to the model of learning with MQs. This intuitive connection will be useful in the proceeding discussion.

Active learning is motivated by the availability of large amounts of unlabeled data at low cost. As such, the typical goal in this model is to develop algorithms with qualitatively improved label complexity (compared to passive learning) at the expense of a larger — but, ideally, still reasonably bounded — set of unlabeled data. Over the past two decades, a large body of work in theoretical machine learning has studied the possibilities and limitations of active learning in a variety of natural and important settings; see, e.g. [FSST97, Das04, Das05, DKM05, BBZ07, BHV10, H+14, HY15, DMRT24**? **].

A prototypical setting where active learning leads to substantial savings is for the task of learning *homogeneous* Linear Threshold Functions (LTFs) or halfspaces. An LTF is any function $h : \mathbb{R}^d \rightarrow \{\pm 1\}$ of the form $h(x) = \text{sign}(w \cdot x + t)$, where $w \in S^{d-1}$ is called the weight vector and $t$ is called the threshold. If $t = 0$, the halfspace is called homogeneous. The problem of learning halfspaces is one the classical problems in machine learning, going back to the Perceptron algorithm [Ros58] and has had a great impact on many other influential techniques, including SVMs [Vap97] and AdaBoost [FS97].

For the class of homogeneous halfspaces under well-behaved distributions (including the Gaussian and isotropic log-concave distributions), prior work has established that $O(d \log(1/\epsilon))$ label queries suffice, where $d$ is the dimension and $\epsilon$ is the desired accuracy [BBZ07, DKM05, BL13]. Moreover, there are computationally efficient algorithms with near-optimal label complexity for this task [ABL17, YZ17, She21], even in the agnostic model that achieve error $O(\text{opt} + \epsilon)$. Unfortunately, this logarithmic dependence on $1/\epsilon$ breaks down for general (potentially biased) halfspaces. Intuitively, this holds because if the bias of a halfspace (the probability mass of the small class) is $p$, then we need to obtain at least $1/p$ labeled examples before we see the first point in the small class. This implies an information-theoretic label complexity lower bound of $\Omega(\min\{1/p, 1/\epsilon\} + d \log(1/\epsilon))$ [Das05], even for realizable PAC learning under the uniform distribution on the sphere. Hanneke et al. [BHV10] showed an information-theoretic label complexity upper bound of $\tilde{O}((1/p)d^{3/2} \log(1/\epsilon))$ for general halfspaces under the uniform distribution on the sphere (via an exponential-time algorithm).

In summary, prior to this work, the possibility that there is an active learner with label complexity $O(d \log(1/\epsilon) + \min\{1/p, 1/\epsilon\})$ and unlabeled sample complexity $\text{poly}(d/\epsilon)$ remained open. Our first main result is an information-theoretic lower bound ruling out this possibility.

**Theorem 1.1** (Main Lower Bound)**.** *For any active learning algorithm $\mathcal{A}$, there is a halfspace $h^*$ that labels $S$ with bias $p$ such that if $\mathcal{A}$ makes less than $\tilde{O}(d/(p \log(m)))$ label queries over $S$, a set of $m$ i.i.d. points drawn from $N(0, I)$, then with probability at least $2/3$ the halfspace $\hat{h}$ output by $\mathcal{A}$ has error more than $p/2$ with respect to $h^*$.*

In particular, if $p$ is chosen as $\Theta(\epsilon \log(1/\epsilon))$, learning a $p$-bias halfspace with error $C\epsilon$ (for any fixed constant $C$) would require a learning algorithm to either make $\tilde{\Omega}(d/\epsilon)$ label queries or have a pool of $2^d$ unlabeled examples. Our information-theoretic lower bound essentially shows that the active setting does not provide non-trivial advantages for the class of general halfspaces, unless the learner is allowed to obtain exponentially many unlabeled examples. (As already mentioned, in this extreme setting, the active learning model approximates PAC learning with MQs.) This motivates the study of learning halfspaces in the stronger model with MQs, where better upper bounds may be attainable.

To circumvent the aforementioned lower bound, we consider the stronger model of PAC learning with MQs. We are interested in understanding the query complexity of learning general halfspaces under the Gaussian distribution. We study this question in the agnostic learning model and establish the following positive result, the proof of which can be found in Appendix G:

**Theorem 1.2** (Main Algorithmic Result)**.** *Consider the problem of agnostic PAC learning halfspaces with membership queries under the Gaussian distribution. There is an algorithm such that for every labeling function $y(x)$ and for every $\epsilon, \delta \in (0, 1)$, it makes $M = \tilde{O}_\delta(\min\{1/p, 1/\epsilon\} + d\,\text{polylog}(1/\epsilon))$* * *memberships queries, runs in $\text{poly}(d, M)$ time, where $p$ is the bias of the optimal halfspace $h^*$, and outputs an $\hat{h} \in H$ such that with probability at least $1 - \delta$, $\text{err}(\hat{h}) \leq O(\text{opt} + \epsilon)$.*

---

*In this paper, we use $O_\delta$ to hide the dependence on $\text{polylog}(1/\delta)$.

In other words, we provide a computationally efficient constant factor agnostic query learner with query complexity $\tilde{O}(\min\{1/p, 1/\epsilon\} + d\,\mathrm{polylog}(1/\epsilon))$. Due to known $d^{\mathrm{poly}(1/\epsilon)}$ complexity lower bounds for achieving optimal error of $\mathrm{opt} + \epsilon$ [DKZ20, DKPZ21, DKR23], the majority of work [DKS18, DKTZ22] in the passive PAC model has focused on designing efficient learners achieving a constant factor approximation of $O(\mathrm{opt}+\epsilon)$These passive learning algorithms have sample complexity $\mathrm{poly}(d, 1/\epsilon)$. Note that, by Theorem 1.1, it is impossible to modify these algorithms (for general halfspaces) to achieve an active learner with low label complexity.

In the realizable setting under the Gaussian distribution, a learner may query many points that are extremely far from the origin to find examples from the small class with few queries. However, such an algorithm is quite fragile to even a tiny amount of noise. In particular, the query complexity achieved by our algorithm establishing Theorem 1.2 is nearly optimal in the agnostic setting.

On the one hand, $\Omega(d\log(1/\epsilon))$ queries are required because describing a halfspace up to error $\epsilon$ requires $d\log(1/\epsilon)$ bits of information [KMT93]. On the other hand, we argue that the overhead term of $\Omega(\min\{1/p, 1/\epsilon\})$ cannot be avoided in the agnostic setting. Such a statement can be deduced from a lower bound of [HKL20]: they showed that in the realizable setting, any algorithm requires at least $\Omega((1/p)^{1-o(1)})$ MQs to see the first example from the small class (where $p$ is the bias of the target halfspace with respect to the uniform distribution on the unit ball); they also showed a similar lower bound of $\Omega(1/p)$ if the underlying distribution is the uniform distribution over the unit sphere. As the dimension $d$ increases, the standard Gaussian distribution is very well approximated by the uniform distribution over a $d$-dimensional sphere with radius $\sim \sqrt{d}$. Thus, an exponentially small level of noise would make every query far from this sphere contain no useful information. This allows us to show that, under the Gaussian distribution with a tiny amount of label noise, $\Omega((1/p)^{1-o(1)})$ queries are needed to see a single example from the small class. The proof of this statement is essentially identical to the argument in [HKL20] for unit ball. The reader is referred to that work for the details.

## 1.1 Preliminaries

For a halfspace $h(x) = \mathrm{sign}(w \cdot x + t)$, $w \in S^{d-1}, t > 0$, we use $p(t) = \mathbf{Pr}_{x\sim N(0,I)}(h(x) = -1)$ to denote its bias. For a halfspace $h(x)$, we define its Chow-Parameter under the standard Gaussian distribution to be $\mathbf{E}_{x\sim N(0,I)}\, xh(x)$. Let $y(x) : \mathbb{R}^d \to \{\pm 1\}$ be a (randomized) labeling function for examples in $\mathbb{R}^d$. We denote by $\mathrm{err}(h) = \mathbf{Pr}_{x\sim N(0,I)}(h(x) \neq y(x))$ to be the error of the hypothesis $h$ and $\mathrm{opt} = \min_{h\in H} \mathrm{err}(h)$, where $H$ is the class of halfspaces over $\mathbb{R}^d$. We will use $h^*$ to denote the halfspace with an error equal to $\mathrm{opt}$. When there is no confusion, we will use $p$ to denote the bias of the optimal halfspace $h^*$.

Let $D_x$ be a distribution over $\mathbb{R}^d$, $y(x)$ be a labeling function over $\mathbb{R}^d$ and $S = \{(x_i, y(x_i))\}_{i=1}^m$ be a set of i.i.d. examples drawn from the distribution $D$ over $\mathbb{R}^d \times \{\pm 1\}$ such that the marginal distribution of $D$ is $D_x$. A membership query takes an $x$ in the support of $D_x$ as input and outputs $y(x)$. A label query takes an $x_i$, where $(x_i, y(x_i)) \in S$ as input and outputs $y(x_i)$. A learning algorithm $\mathcal{A}$ is allowed to use membership queries/label queries and aims to output a halfspace hypothesis $\hat{h}$ such that $\mathrm{err}(\hat{h}) \leq O(\mathrm{opt} + \epsilon)$ by making as few queries as possible.

# 2 Nearly-Tight Lower Bound on Label Complexity: Proof of Theorem 1.1

In this section, we prove our information-theoretic lower bound on the label complexity of active learning general halfspaces under the Gaussian distribution.

Before presenting our proof, we provide high-level intuition behind Theorem 1.1 and the strategy of our proof. Previous work, see, e.g. [Das04, DKM05, HKL20], showed that if $S$ is a set of examples drawn uniformly from the unit sphere, and if $h^*$ is a halfspace with bias $p$ that is chosen uniformly, the following holds: no matter which query strategy a learning algorithm $\mathcal{A}$ uses, for the first $r$ queries, in expectation only $pr$ of them fall into the small cap on the sphere cut by $h^*$. Thus, if $\mathcal{A}$ makes less than $1/(2p)$ queries, it will with constant probability not see any negative examples; and it is therefore impossible to learn the target halfspace.

In the Gaussian case, we will use a similar but stronger idea. If we are able to learn $h^*$ up to error $p/2$ with few queries, then we can randomly partition $S$ into two sets, use the first set to learn the halfspace and use the second part to find $d$ negative examples by paying another $O(d)$ queries in expectation. Formally, we have the following statement, the proof of which can be found in Appendix B.1.

**Lemma 2.1.** *Suppose there is an active learning algorithm that can make $r$ label queries over a pool $S$ of $m \geq \mathrm{poly}(d/p)$ examples drawn from $N(0, I)$ and learn any halfspace $h^*(x) = \mathrm{sign}(w^* \cdot x + t^*)$ with bias $p$ up to error $p/2$ with probability at least $2/3$. Then there is an algorithm such that given a pool of $2m$ random examples $S$ drawn from the standard Gaussian distribution with hidden labels by some halfspace $h^*(x) = \mathrm{sign}(w^* \cdot x + t^*)$ with bias $p$, it makes $r + O(d)$ queries and finds $d$ negative examples from $S$ with probability $1/2$.*

We will show that finding $d$ negative examples from $S$ requires many queries. The idea is that since $S$ is sampled from a standard Gaussian in high dimensions, every pair of examples is almost orthogonal unless $m$ is as large as $2^d$. If we have made $1/\epsilon$ queries over $S$ and found our first negative example, then this negative example will only provide us with very little knowledge to find the next negative example — as no example in the pool has a large correlation with it. Therefore, it will still take us another approximately $1/\epsilon$ queries to find the next negative example. Such an issue only disappears after we have already found roughly $d$ negative examples; at which time, the average of the $d$ examples has a good correlation with $w^*$. Therefore, it would take us roughly $d/\epsilon$ queries in total. We remark that such an argument is hard to formalize, because, besides negative examples, the algorithm has also seen many positive examples in the process. It is thus challenging to argue that the algorithm cannot make good use of the information obtained from these positive examples.

To overcome this difficulty, our proof strategy works as follows. Each algorithm $\mathcal{A}$ can be described as a decision tree. Each tree node represents the example queried in a given round. Every time the algorithm sees a negative example, it moves to the left; otherwise, it moves to the right. Suppose that $\mathcal{A}$ wants to find $k$ negative examples with $r$ queries. Then there are at most $\binom{r}{k} \leq (er/k)^k$ paths of the tree, where $\mathcal{A}$ successfully finds $k$ negative examples, and for each of the paths there are exactly $k$ examples that are negative upon queried. For a $k$-tuple of examples, we will derive a deterministic condition such that if the $k$ examples satisfy the condition, a random halfspace with bias $p$ will have only roughly $p^k$ probability to label all of the $k$ examples negative.

Formally, we establish the following technical lemma (see Appendix B.2 for the proof).

**Lemma 2.2.** *Let $A \in \mathbb{R}^{k \times d}$ be a matrix with row vectors $x_1, \ldots, x_k$. Let $t^* > C > 0$ for some sufficiently large constant $C$. Let $h^*(x) = \mathrm{sign}(w^* \cdot x + t^*)$ be a random halfspace with bias $p$ with $w^* \sim S^{d-1}$ chosen uniformly from $S^{d-1}$. If $\left\| AA^\top - dI \right\|_2 \leq O(d/(t^*)^2)$, then with probability at most $O(p \log(1/p))^k$, where $p$ is the bias of $h^*$ under the Gaussian distribution, $h^*(x_i) = -1$ for $i = 1, \ldots, k$.*

Thus, if the $\binom{r}{k}$ tuples all satisfy such a condition, then $\mathcal{A}$ will succeed with a fairly tiny probability unless $r$ is larger than $k/p$. So, in the last step of the proof, we will show in Lemma 2.3 that by taking $k \approx d/(\log(m)\mathrm{polylog}(1/p))$, with high probability every $k$-tuple of examples in $S$ will satisfy the deterministic condition. Thus, no algorithm can succeed with a constant probability, unless it makes $\tilde{\Omega}(d/(p \log(m)))$ queries. The proof of Lemma 2.3 can be found in Appendix B.3.

**Lemma 2.3.** *Let $S \subseteq \mathbb{R}^d$ be a set of $m$ examples drawn i.i.d. from $N(0, I)$. Let $t^* > C > 0$ for a sufficiently large constant $C$ and $k = O(d/\log(m)(t^*)^4)$. Then, with probability at least $2/3$, for every $k$-tuple of examples $\{x_1, \ldots, x_k\} \subseteq S$, $\left\| AA^\top - dI \right\|_2 \leq d/(t^*)^2$, where $A \in \mathbb{R}^{k \times d}$ be a matrix with row vectors $x_1, \ldots, x_k$.*

*Proof of Theorem 1.1.* We will start by showing that, given a set $S$ of $m$ points drawn i.i.d. from a Gaussian distribution, the following holds. With probability at least $2/3$, for every algorithm $\mathcal{A}$ there exists a halfspace $h^* = \mathrm{sign}(w^* \cdot x + t^*)$ with bias $p$ such that if $\mathcal{A}$ makes only $r = \tilde{O}(d/p \log(m))$ label queries over $S$, then with probability at least $2/3$ it will not be able to find $k$ negative examples in $S$ for some $k \leq d$. By Yao's minimax principle, it is sufficient to show that there is a distribution over halfspaces $\bar{h}^*$ such that for any deterministic active learning algorithm, the following holds: given $m$ random Gaussian examples, if the learning algorithm makes $r$ queries, with probability $2/3$ it cannot find $k$ negative examples. We will fix the threshold $t^*$ of $h^*$ and draw $w^*$ uniformly from the unit sphere.

By Lemma 2.3, we know that by choosing $k = O(d/\log(m)(t^*)^4)$, with probability at least $2/3$, for every $k$-tuple of examples $x_1, \ldots, x_k \in S$, $\left\| AA^\top - dI \right\|_2 \leq d/(t^*)^2$, where $A \in \mathbb{R}^{k \times d}$ is a matrix with row vectors $x_1, \ldots, x_k$. By Lemma 2.2, we know that every $k$-tuple of examples $x_1, \ldots, x_k \in S$ has a probability $\alpha^k$, which is at most $O(p \log p)^k$ to be labeled all negative by the random halfspace $h^*$. Notice that every query algorithm can be expressed as a binary tree $T$. Each node of the tree represents an example where the algorithm makes queries at a time. If the example at node $v$ is negative, then the algorithm will query the left child of $v$, and otherwise it will query the right child of $v$. The algorithm stops making queries when either it has queried $r$ examples or it has queried $k$ negative examples. In particular, for a given search algorithm, there are at most $\binom{r}{k}$ different possible outcomes where it successfully finds $k$ negative examples. Furthermore, for each of the possible outcomes, there is a set of $k$ examples in $S$ that correspond to the $k$ negative examples the algorithm finds. Thus, the probability that the algorithm successfully finds $k$ negative examples is bounded above by the probability that there exists one of the $\binom{r}{k}$ $k$-tuples of examples in $S$ that are all labeled negative by $h^*$. Such a probability can be bounded above by

$$\binom{r}{k}\alpha^k \leq \left( \frac{er}{k} O(p\log(1/p)) \right)^k \leq 2/3 \ ,$$

if $r \leq O(k/p\log(1/p)) = O(d/(p\log(m)\text{polylog}(1/p)))$. By Lemma 2.1, we know that if we can make $O(d/(p\log(m)\text{polylog}(1/p)))$ label queries to learn a $p$-biased halfspace up to error $p/2$ over a set $S$ of $m/2$ Gaussian examples, then we can use $O(d/(p\log(m)\text{polylog}(1/p)))$ queries to find $d$ negative examples among $m$ Gaussian points. This leads to a contradiction. Thus, the label complexity of the learning problem is $\tilde{\Omega}(d/(p\log(m)))$, as desired.

$\square$

## 3 Robust Learning of General Halfspaces with MQs: Proof of Theorem 1.2

In this section, we present our main algorithmic result, Theorem 1.2. We refer the readers to Appendix G for the full proof of Theorem 1.2. Throughout the paper, we will assume for convenience that the noise level opt $\leq \epsilon$. Such an assumption can be made without loss of generality, as discussed in Appendix C.1. We first present our main algorithm, Algorithm 1. Algorithm 1 will maintain a list of $\text{polylog}(1/\epsilon)$ candidate hypotheses at least one of which has error $O(\text{opt} + \epsilon)$. We will then use a standard tournament approach to find an accurate hypothesis among them.

---

**Algorithm 1** QUERY LEARNING HALFSPACE(Efficient Agnostic Learning Halfspaces with Queries)

---

**Input:** error parameter $\epsilon \in (0, 1)$, confidence parameter $\delta \in (0, 1)$
**Output:** halspace $\hat{h}(x) = \text{sign}(\hat{w} \cdot x + \hat{t})$, where $\hat{w} \in S^{d-1}, \hat{t} > 0$
$\mathcal{C} \leftarrow \emptyset$             ▷ **Create a list of candidate hypothesises** $\mathcal{C}$
Use $\tilde{O}(\min\{1/p, 1/\epsilon\})$ queries to estimate $p$ by some $\hat{p}$ such that $\hat{p} \leq p \leq 2\hat{p}$ (or verify $p < C\epsilon$ and return $+1$, the constant hypothesis).
Let $t_a, t_b > 0$ such that a halfspace with threshold $t_a$ has bias $2\hat{p}$ and with threshold $t_b$ has bias $\hat{p}$. Build grid points $t_a = t_0 < t_1 < \cdots < t_\psi = t_b$ such that $|t_{i+1} - t_i| = 1/(2\log(1/\epsilon)), \forall i \leq \psi - 1$.
    ▷ **Guess the true threshold** $t^*$ **with** $t' \in \{t_0, t_1, \ldots\}$
**for** $j = 0, \ldots, \psi$ **do**
    Repeat the following procedure $\text{polylog}(1/\epsilon)\log(1/\delta)$ times
    $w_0 \leftarrow \text{INITIALIZATION}(\epsilon, t_j, \delta/\text{polylog}(1/\epsilon))$     ▷ **Find a** $w_0 \in S^{d-1}$ **as a warm start**
    $(w_T, \hat{t}) \leftarrow \text{REFINE}(w_0, t_j, \epsilon.\delta/\text{polylog}(1/\epsilon))$ ▷ **Find a** $w_T \in S^{d-1}$ **close enough to** $w^*$ **and** $\hat{t}$ **close enough to** $t^*$ **based on** $w_0$
    $\mathcal{C} \leftarrow \mathcal{C} \cup \{\text{sign}(w_T \cdot x + \hat{t})\}$     ▷ **Add a new candidate hypothesis to** $\mathcal{C}$
Find a good hypothesis $\hat{h}$ from $\mathcal{C}$ using Lemma C.1, a standard tournament approach
**return** $\hat{h}$

---

At the beginning of Algorithm 1, we will use random queries to approximately estimate the bias $p$ of the optimal halfspace up to a constant factor. As we will discuss in Appendix C.2, such an estimation can be done with only $\tilde{O}(\min\{1/p, 1/\epsilon\})$ queries by applying a doubling trick to the coin estimation problem. In particular, if we find $p < C\epsilon$, we can directly output a constant hypothesis as it has

error only $O(\epsilon)$. Since $t^*$ is unknown to us, such an approach can prevent us from using some $t'$ which is much larger than $t^*$ in the rest of the learning procedure, which will potentially lead to a larger query complexity. With such a $\hat{p}$, $t^*$ will fall into a reasonable range $[t_a, t_b]$. We next partition $[t_a, t_b]$ into a grid of size $O(1/\log(1/\epsilon))$ and use each of the grid points as an initial guess of $t^*$. In particular, at least one of these grid points $t_j$ is $O(1/\log(1/\epsilon))$ close to $t^*$. Although such a $t_j$ is not accurate enough to be used in the final output hypothesis, as $t^* \leq \sqrt{\log(1/\epsilon)}$, we will show later that such a $t_j$ is enough for us to use it to learn $w^*, t^*$ accurately. Suppose now we have such a good $t_j$. We will design two subroutines that make use of $t_j$ to produce a good hypothesis $\text{sign}(w_T \cdot x + \hat{t})$. The first algorithm will take $t_j$ and the noise level $\epsilon$ as its input and produce a unit vector $w_0$ as an initialization. We will show in Section 3.2 that as long as $|t_j - t^*| \leq 1/\log(1/\epsilon)$, we can with probability at least $1/\log(1/\epsilon)$ produce some $w_0$ such that $\theta(w_0, w^*) \leq O(1/t_j)$. By repeating such an initialization algorithm $\text{polylog}(1/\epsilon)$ times, with high probability one of these runs will succeed. In particular, such an algorithm has a query complexity of $\tilde{O}(1/p + d\,\text{polylog}(1/\epsilon))$. Now assume we have such a $w_0$ as a warm-start. Our second subroutine is to refine the direction $w_0$ and the threshold $t_j$. More specifically, we will maintain a unit vector $w_i$ such that $\theta_i = \theta(w_i, w^*)$ and an upper bound $\sigma_i$ for $\sin(\theta_i/2)$. In each round of the refining algorithm, we will use $\tilde{O}(d)$ queries to update $w_i$. In particular, in each round $\sigma_i$ will decrease by a constant factor and thus after at most $T = \tilde{O}(\log(1/\epsilon))$ rounds, we will have $\sin(\theta_T/2) \leq \sigma_T = C\epsilon \exp(t_j^2/2)$. As we will show in Section 3.1, provided the correct $t^*$, $\text{sign}(w_T \cdot x + t^*)$ is at most $O(\epsilon)$ far from $h^*$. However, to output a good hypothesis, we still need to learn $t^*$ up to a high accuracy. When $t^*$ is small, we even have to estimate $t^*$ up to error $O(\epsilon)$, which typically needs many queries. However, as we will show in Section 3.1, given $w_T$ close enough to $w^*$, we are able to combine the localization technique used in [DKS18] with this fact to learn $t^*$ using only $O(\log(1/\epsilon))$ queries. This gives an overview of Algorithm 1 and its query complexity.

## 3.1 Refining A Warm-Start

We will start by discussing how to refine a warm start $w_0$ by proving the following theorem. The proof of the theorem and the main algorithm, Algorithm 3 can be found in Appendix D.5.

**Theorem 3.1.** *Let* $h^*(x) = \text{sign}(w^* \cdot x + t^*)$ *be a halfspace such that* $\text{err}(h^*) = \text{opt} \leq \epsilon$. *Let* $t' \leq \sqrt{\log(1/\epsilon)}, w_0 \in S^{d-1}$ *be inputs of Algorithm 3. If* $t' - 1/\log(1/\epsilon) \leq t^* \leq t'$, $t' \exp((t')^2/2) \leq 1/(C\epsilon)$ *and* $\sin(\theta(w_0, w^*)/2) \leq \sigma_0 := \min\{1/t', 1/2\}$, *then Algorithm 3 makes* $M = \tilde{O}_\delta(d\,\text{polylog}(1/\epsilon))$ *membership queries, runs in* $\text{poly}(d, M)$ *time, and outputs* $(w_T, \hat{t})$ *such that with probability at least* $1 - O(\delta)$, $\text{err}(\text{sign}(w_T \cdot x + \hat{t})) \leq O(\epsilon)$.

As we discussed in Section 3, we will assume we have some $t'$ such that $t' - 1/\log(1/\epsilon) \leq t^* \leq t'$ and some $w_0$ such that $\sin(\theta_0/2) \leq \sigma_0 = \min\{1/t', 1/2\}$, i.e., some initial knowledge of $t^*, w^*$. Our algorithm runs in iterations and will maintain some $w_i$ in round $i$. We will maintain some unit vector $w_i$ and use $\|w_i - w^*\| = 2\sin(\theta_i/2)$ to measure the progress made by Algorithm 3. The method we use to update $w_i$ is a simple projected gradient descent algorithm. Specifically, we will construct a random vector $G_i$ over $\mathbb{R}^d$ such that $G_i \perp w_i$ and in expectation $g_i = \mathbf{E}\,G_i$ has bounded length and a good correlation with respect to $w^*$. We will show in the following lemma that by estimating $\mathbf{E}\,G_i$ up to constant error with $\hat{g}_i$ and using the update rule $w_{i+1} = \text{proj}_{S^{d-1}}(w_i + \mu_i \hat{g}_i)$, we are able to significantly decrease $\theta_i$. The proof of Lemma 3.2 can be found in Appendix D.1.

**Lemma 3.2.** *Let* $w^*, w_i \in S^{d-1}$ *such that* $w^* = a_i w_i + b_i u$, *where* $u \in S^{d-1}, u \perp w_i, a_i, b_i > 0, a_i^2 + b_i^2 = 1$. *Let* $\theta_i = \theta(w_i, w^*)$. *Let* $G_i$ *be a random vector drawn from some distribution* $\mathcal{D}$ *such that with probability* 1, $G_i \perp w_i$. *Let* $g_i$ *be the mean of* $G_i$. *Let* $\hat{g}_i$ *be the empirical mean of* $G_i$ *and* $\mu_i > 0$. *The update rule* $w_{i+1} = \text{proj}_{S^{d-1}}(w_i + \mu_i \hat{g}_i)$ *satisfies the following property,*

$$\|w_{i+1} - w^*\|^2 \leq \|w_i - w^*\|^2 - 2\mu_i b_i g_i \cdot u + 2\mu_i b_i \|\hat{g}_i - g_i\| + \mu_i^2 \|\hat{g}_i^2\|.$$

*Furthermore, if* $\sin(\theta_i/2) \leq \sigma_i \in (0,1)$ *and there exist constant* $c_1, c_2$ *such that* $g_i \cdot u \geq c_1/10$, $\|\hat{g}_i\| \leq c_1$ *and* $\|g_i - \hat{g}_i\| \leq c_2 \leq c_1/40$, *then there exist constant* $C_1, C_2 > 8$ *such that by taking* $\mu_i = \sigma_i/C_1$ *and* $\sigma_{i+1} = (1 - 1/C_2)\sigma_i$, *it holds that* $\sin(\theta_{i+1}/2) \leq \sigma_{i+1}$. *In particular, if* $\sin(\theta_i/2) \leq 3\sigma_i/4$ *and* $\|\hat{g}_i\| \leq c_1$ *then* $\sin(\theta_{i+1}/2) \leq \sigma_{i+1}$ *always holds.*

In the rest of the section, we will show that as long as $w_i$ is not good enough, we can always efficiently construct a random vector $G_i$ whose expectation points to the correct direction and we can use very

few queries to estimate its expectation up to a desired accuracy. We adapt the localization technique used in [DKS18] to achieve this goal.

### 3.1.1 Finding A Good Gradient via Localization

In the $i$-th round of Algorithm 3, we write $w^* = a_i w_i + b_i u_i$, where $u_i \in S^{d-1}, u_i \perp w_i, a_i, b_i > 0, a_i^2 + b_i^2 = 1$. Recall that $\sigma_i$ is an upper bound we maintain for $\sin(\theta_i/2)$. We will construct the random gradient as follows

$$G_i := \mathrm{proj}_{w_i^\perp} zy(A_i^{1/2} z - \tilde{t} w_i),$$

where $z \sim N(0, I)$, $A_i = I - (1 - \sigma_i^2) w_i w_i^t$ and $\tilde{t} \in (0, t')$ is a scalar. To see why $G_i$ is a good choice, we will start by analyzing $G_i$ assuming the noise rate $\mathrm{opt} = 0$. To simplify the notation, denote by $\ell_i(z) = \mathrm{sign}((a_i w_i + b_i u_i/\sigma_i)z + (t^* - a\tilde{t})/\sigma_i)$ and $\bar{g}_i = \mathbf{E}_{z \in N(0,I)} \mathrm{proj}_{w_i^\perp} z\ell_i(z)$. A simple calculation gives us the following result.

**Fact 3.3.** *Let $h(x) = \mathrm{sign}(w \cdot x + t)$ be a halfspace. Let $v \in S^{d-1}$ such that $w = av + bu$, where $a, b > 0, a^2 + b^2 = 1$, $u \in S^{d-1}, u \perp v$. Let $s, \sigma > 0$ be real numbers and define $A = I - (1 - \sigma^2)vv^t$. For each $z \in \mathbb{R}^d$, define $\tilde{z} := A^{1/2} z - sv$. Then $h(\tilde{z}) = \ell(z)$, where $\ell$ is the following halfspace*

$$\ell(z) = \mathrm{sign}((av + bu/\sigma) \cdot z + (t - as)/\sigma) .$$

Fact 3.3 implies that if $\mathrm{opt} = 0$, then it always holds that $f_i(z) := y(A_i^{1/2} z - \tilde{t} w_i) = \ell_i(z), \forall z \in \mathbb{R}^d$ and we can view $z$ as examples labeled by a halfspace $\ell_i(z)$. In particular, $\mathbf{E}_{z \sim N(0,I)} zf_i(z)$ is the Chow-Parameter of the halfspace $\ell_i(z)$ under the standard Gaussian distribution.

**Fact 3.4** (Lemma C.3 in [DKS18]). *Let $h(x) = \mathrm{sign}(w \cdot x + t)$, where $w \in S^{d-1}$ be a halfspace. Then $\mathbf{E}_{z \sim N(0,I)} zh(z) = \sqrt{\frac{2}{\pi}} \exp\left(-t^2/2\right)w$.*

By Fact 3.4, in the noiseless case, $\mathbf{E}_{z \sim N(0,I)} zf_i(z)$ is parallel to $(a_i w_i + b_i u_i/\sigma_i)$ with length $\Theta(\exp(-T_i^2))$, where $T_i = \frac{t^* - a_i \tilde{t}}{\sigma_i \sqrt{a_i^2 + b_i^2/\sigma_i^2}}$ and $g_i = \bar{g}_i$ is exactly the $u_i$ component of the Chow-Parameter. In particular, if $T_i$ is constant, then by estimating $g_i$ using $\hat{g}_i$ up to a small constant error using $\tilde{O}(d)$ queries, we are able to use Lemma 3.2 to improve $w_i$. Assuming we set $\tilde{t} = t^*$, as $\sigma_i t' \leq 1$ and $b_i \leq O(\sigma_i)$, it is easy to check $T_i$ can be bounded by some universal constant. However, as we mentioned before, we only know $|t' - t^*| \leq \frac{1}{\log(1/\epsilon)}$, when $w_i$ getting close to $w^*$, $\sigma_i$ could become very small and an error of $1/\log(1/\epsilon)$ could potentially blow up $T_i$, making the signal we want quite small. Such an issue is problematic for the algorithm, especially when $f_i(z)$ is a noisy version of $\ell_i(z)$. To overcome such an issue, we prove the following structural lemma in Appendix D.2 showing that we can always check whether the choice of $\tilde{t}$ is good or not, by looking at the bias of $\ell(z)$, using $\tilde{O}(1)$ queries. Using this method, we can perform a binary search for $\tilde{t}$ to find a correct choice in at most $\log(1/\epsilon)$ rounds. Furthermore, as long as we select the correct $\tilde{t}$, it must hold that $|\tilde{t} - t^*| \leq O(\sigma_i)$. In particular, as $\sigma_T = C\epsilon \exp((t')^2/2)$, such a $\tilde{t}$ is a good enough estimate for $t^*$ to be used in the final hypothesis.

**Lemma 3.5.** *Let $w^*, w_i \in S^{d-1}$ such that $w^* = a_i w_i + b_i u_i$, where $u_i \in S^{d-1}, u \perp w_i, a_i, b_i > 0, a_i^2 + b_i^2 = 1$. Let $t^*, t', \sigma_i, \epsilon$ be positive real numbers such that $0 \leq t^* \leq t', \sin(\theta_i/2) \leq \sigma_i$, and $\sigma_i t' \leq 1$. Define $T_i := \frac{t^* - a_i \tilde{t}}{\sigma_i \sqrt{a_i^2 + b_i^2/\sigma_i^2}}$, $\ell_i(z) = \mathrm{sign}((a_i w_i + b_i u_i/\sigma_i)z + (t^* - a\tilde{t})/\sigma_i)$ and $\bar{g}_i = \mathbf{E}_{z \in N(0,I)} \mathrm{proj}_{w_i^\perp} z\ell_i(z)$ for some $\tilde{t} \in [0, t']$. Then the following three properties hold.*

1. *There exists an interval $I_{t'} \subseteq [0, t']$ of length at least $\sigma_i$ such that for every $\tilde{t} \in I_{t'}, |T_i| \leq 5$.*

2. *When $|T_i| \leq 6$, it holds that $\bar{g}_i \cdot u_i = \|\bar{g}_i\|$ and $e^{-19} b_i/\sigma_i \leq \|\bar{g}_i\| \leq 2e^{-19}$.*

3. *For every $\left|\tilde{t} - t^*\right| > 40\sigma_i$ and $\tilde{t} < t', |T_i| > 10$.*

### 3.1.2 Robustness Analysis

So far, we have only considered the case when $\mathrm{opt} = 0$. Due to the presence of noise, it is impossible for us to estimate $\bar{g}_i = \mathbf{E}_{z \in N(0,I)} \mathrm{proj}_{w_i^\perp} z\ell_i(z)$ because we only have a noisy version $f_i(z)$ of

$\ell_i(z)$. In this section, we will show that as long as $w_i$ is close to $w^*$ and $|t' - t^*| \leq 1/\log(1/\epsilon)$, the probability that for a Gaussian point $z$, $\ell_i(z) \neq f_i(z)$ is at most a tiny constant. This is incomparable with the bias of $\ell_z(z)$ if $\tilde{t}$ is chosen correctly, and does not affect the algorithm too much. We start with the following lemma which bounds the probability of $\ell_i(z) \neq f_i(z)$.

**Lemma 3.6.** *Let* $h^*(x) = \text{sign}(w^* \cdot x + t^*)$ *be a halfspace such that* $\text{err}(h^*) = \text{opt} \leq \epsilon$. *Let* $\tilde{t}, \sigma_i, t'$ *be real numbers such that* $\tilde{t} \leq t'$ *and* $\sigma_i t' \leq 1, \sigma_i \leq 1/2$. *Let* $w^* = a_i w_i + b_i u_i$, *where* $u_i \in S^{d-1}, u \perp w_i, a_i, b_i > 0, a_i^2 + b_i^2 = 1$. *Define* $\ell_i(z) = \text{sign}((a_i w_i + b_i u_i/\sigma_i)z + (t^* - a\tilde{t})/\sigma_i)$ *and* $f_i(z) = y(A_i^{1/2} z - \tilde{t} w_i)$. *Then* $\mathbf{Pr}_{z \sim N(0,I)}(\ell_i(z) \neq f_i(z)) \leq \epsilon \exp(\tilde{t}^2/2 + 4)/\sigma_i$. *In particular, if* $\sigma_i \geq C \exp((t')^2/2)\epsilon$, *for some sufficient large constant* $C$, *then there is a sufficiently small constant* $c$ *such that* $\mathbf{Pr}_{z \sim N(0,I)}(\ell_i(z) \neq f_i(z)) \leq c \leq e^{-40}$.

The proof of Lemma 3.6 leverages the $(v, s, \sigma)$- rejection procedure introduced in [DKS18] (see Appendix D.3). We will use Lemma 3.6 to analyze the gradient descent approach we described in the presence of noise. Formally, we establish the following lemma (see Appendix D.4 for the proof).

**Lemma 3.7.** *Let* $w^*, w_i \in S^{d-1}$ *such that* $w^* = a_i w_i + b_i u_i$, *where* $u_i \in S^{d-1}, u \perp w_i, a_i, b_i > 0, a_i^2 + b_i^2 = 1$. *Let* $t^*, t', \sigma_i, \epsilon$ *be positive real numbers such that* $0 \leq t^* \leq t'$, $\sin(\theta_i/2) \leq \sigma_i$, $\sigma_i \geq C \exp((t')^2/2)\epsilon$, *and* $\sigma_i t' \leq 1$. *Let* $h^*(x) = \text{sign}(w^* \cdot x + t^*)$ *be a halfspace such that* $\text{err}(h^*) = \text{opt} \leq \epsilon$. *Define* $T_i := \frac{t^* - a_i \tilde{t}}{\sigma_i \sqrt{a_i^2 + b_i^2/\sigma_i^2}}$, $\ell_i(z) = \text{sign}((a_i w_i + b_i u_i/\sigma_i)z + (t^* - a\tilde{t})/\sigma_i)$,

$\bar{g}_i = \mathbf{E}_{z \in N(0,I)} \text{proj}_{w_i^\perp} z \ell_i(z)$ *and* $g_i = \mathbf{E}_{z \in N(0,I)} \text{proj}_{w_i^\perp} z f_i(z)$, *where* $f_i(z) = y(A_i^{1/2} z - \tilde{t} w_i)$ *for some* $\tilde{t} \in [0, t']$. *Let* $\eta_i := \mathbf{Pr}_{z \sim N(0,I)}(\ell_i(z) \neq f_i(z))$ *and* $p_i$ *be the probability that* $f_i(z) = -1$. *Then the following two properties hold.*

1. *If* $p_i \in (e^{-18}, 1 - e^{-18})$, *then* $|T_i| < 6$ *and if* $|T_i| < 5$, *then* $p_i \in (e^{-16}, 1 - e^{-16})$.

2. $g_i \cdot u_i \geq \bar{g}_i \cdot u_i - 2\sqrt{e}\eta_i \sqrt{\log(1/\eta_i)}$ *and* $\|g_i\| \leq \|\bar{g}_i\| + 2\sqrt{e}\eta_i \sqrt{\log(1/\eta_i)}$.

Lemma 3.7 says as the noise level is small, it will not affect the structure lemma we established in Lemma 3.5 too much, and thus we are able to find the correct threshold $\tilde{t}$ by checking the probability of $f_i(z) = -1$. Furthermore, as long as we choose the correct threshold $\tilde{t}$, $g_i$, the noisy version of $\bar{g}_i$ still satisfies the conditions in the statement of Lemma 3.2 and thus can be used to improve $w_i$.

## 3.2 Finding A Good Initialization

In Section 3.1, we have shown that given some $w_0$ non-trivially close to $w^*$ and some $t'$ such that $t' - \frac{1}{\log(1/\epsilon)} \leq t^* \leq t'$, we can use Algorithm 3 to learn a good hypothesis with high probability. In this section, we show how to find such a good initialization $w_0$ using a few membership queries. The most common way to get such a warm-start is by robustly estimating the Chow-Parameter (see for example [She21, YZ17]) using Fact 3.4. Such an approach does not work for general halfspaces because the length of the length of the Chow-Parameter can be as small as $\tilde{O}(p)$, and thus needs roughly $d/p$ random queries to estimate. In this section, we show how to overcome such an issue using a label smoothing technique, which has been useful in related problems [DKK+23]. The main results in this step can be summarized as follows. The proof of Theorem 3.8 is deferred to Appendix E.2

**Theorem 3.8.** *Let* $h^*(x) = \text{sign}(w^* \cdot x + t^*)$ *and* $y(x)$ *be any labeling function such that* $\text{err}(h^*) = \text{opt} \leq \epsilon \leq 1/C$ *for some large enough constant* $C$. *If* $|t - t^*| \leq 1/\log(1/\epsilon)$, *then with probability at least* $1/3$, *Algorithm 2 makes* $M = \tilde{O}(1/p + d\log(1/\epsilon))$, *runs in* $\text{poly}(d, M)$ *time, and outputs some* $w_0$ *such that* $\sin(\theta(w_0, w^*)/2) \leq \max\{\min\{1/t, 1/2\}, O(\eta\sqrt{\log(1/\eta)})\}$, *where* $\eta = \epsilon/p$.

Due to the space limitations, here we only consider the case when $t^*$ is not extremely large, which roughly covers the regime when $\eta\sqrt{\log(1/\eta)} \leq 1/t$. This suffices to capture some of the ideas and illustrate the power of the smoothed labeling. For the case when $\eta\sqrt{\log(1/\eta)} > 1/t$, we are still able to find such a warm start by leveraging the smoothed label method in combination with the technique used in Section 3.1 in a more complicated way. We postpone this analysis to Appendix F. Our algorithm, Algorithm 2, to find a warm start is presented as follows.

To analyze Algorithm 2, we introduce the following definitions and notations.

---
**Algorithm 2** INITIALIZATION 1(Finding a good initialization under unextreme threshold)
---
**Input:** error parameter $\epsilon \in (0,1)$, confidence parameter $\delta \in (0,1)$, threshold $t > 0$
**Output:** $w_0 \in S^{d-1}$
Keep some $x \sim N(0, I)$ and query $y(x)$ until see some $x_0$ such that $y(x_0) = -1$
**for** $i = 1, \ldots, m = \tilde{O}(d \log(1/\epsilon))$ **do**
    Sample $z_i \sim N(0, I)$ and query $\tilde{y}(x_0^{(i)}) := y(\sqrt{1 - \rho^2} x_0 + \rho z_i)$ with $\rho := \min\{1/t, 1\}$
Let $u_0 := \frac{1}{m} \sum_{i=1}^{m} z_i \tilde{y}(x_0^{(i)})$
**return** $w_0 := u_0 / \|u_0\|$

---

**Definition 3.9** (Smoothed Label). *Let $x \in \mathbb{R}^d$ be a point and $y(x)$ be any labeling function. For $\rho \in [0, 1]$, define the random variable $\tilde{x} = \sqrt{1 - \rho^2} x + \rho z$, where $z \sim N(0, I)$. The smoothed label of $x$ with parameter $\rho$ is defined as $\tilde{y}(x) := y(\tilde{x})$.*

We will require the following fact (whose proof follows via a direct calculation):

**Fact 3.10.** *Let $h^*(x) = \text{sign}(w^* \cdot x + t^*)$ be a halfspace. Let $x, z \in \mathbb{R}^d$ and define $\tilde{x} := \sqrt{1 - \rho^2} x + \rho z$. Then $\tilde{h}(z) := h^*(\tilde{x}) = \text{sign}(w^* \cdot z + (t^* + \sqrt{1 - \rho^2} w^* \cdot x)/\rho)$ is another halfspace for $z$ with threshold $(t^* + \sqrt{1 - \rho^2} w^* \cdot x)/\rho$.*

Let $h^* = \text{sign}(w^* \cdot x + t^*)$ be an optimal halfspace and let $y(x)$ be any labeling function such that $\text{err}(h^*) = \text{opt} \leq \epsilon$. For $x \in \mathbb{R}^d$, we denote by $\eta(x) := \mathbf{Pr}(h^*(\tilde{x}) \neq \tilde{y}(x))$, the noise level of the smoothed label. Assuming that we are given a random negative example $x_0$, then with constant probability, it is close to the decision boundary, i.e., $w^* \cdot x_0 \in (-t^* - \frac{1}{t^*}, -t^*)$. This implies that the threshold of $\bar{h}$, the halfspace corresponding to the smoothed label at $x_0$, is between $(-1, 1)$. Moreover, the Chow-Parameter of $\bar{h}$ under the standard Gaussian distribution is parallel to $w^*$ with a constant length, by Fact 3.4. If $\text{opt} = 0$, then for every $t \leq \sqrt{\log(1/\epsilon)}$, we only need another $\tilde{O}(d \log(1/\epsilon))$ queries to estimate the Chow-Parameter of $\bar{h}$ up to error $O(1/t)$; thus, we get a warm start $w_0$ such that $\sin(\theta_0/2) \leq 1/t$, given $|t - t^*|$ is small. Therefore, the total number of queries we use to run Algorithm 2 is $\tilde{O}(1/p + d \log(1/p))$. However, in general, it is impossible to estimate $w^*$ up to arbitrary accuracy — even using an infinite number of queries — because of the presence of noise. In fact, using a random $x_0$ is important for Algorithm 2 to succeed. If we are given some adversarially selected $x_0$, even if it is close to the decision boundary, the above method can easily fail. This is because almost all the queries we made are in a small neighborhood of $x_0$ and could be corrupted by noise arbitrarily. However, we show in Appendix E.1 that, with a probability at least $2/3$, the noise level $\eta(x_0)$ of the smoothed label around $x_0$ is at most $O(\epsilon/p)$, if $x_0$ is a random example given $y(x_0) = -1$; and thus we can still estimate $w^*$ to a desired accuracy provided $\epsilon/p$ is not too large.

**Lemma 3.11.** *Let $h^*(x) = \text{sign}(w^* \cdot x + t^*)$ be a halfspace and $y(x)$ be any labeling function such that $\text{err}(h^*) = \text{opt} \leq \epsilon$. Let $x \sim N(0, I)$ conditioned on $y(x) = -1$ be a Gaussian example with a negative label. If $p > C\epsilon$ for some large enough constant $C$, then with probability at least $1/2$ we have $\eta(x) \leq 5\epsilon/p$ and $w^* \cdot x \in (-t^* - 1/t^*, -t^*)$.*

Finally, we briefly discuss how to obtain a warm start when the threshold $t^*$ is very large. The details of this method can be found in Appendix F. By Theorem 3.8, when $p$ is small, we are only able to get some $w_0$ such that $\sin(\theta(w_0, w^*)) \leq O(\eta \sqrt{\log(1/\eta)})$ for $\eta = \epsilon/p$. One possible approach is to use the localization technique we use in Section 3.1 to refine such $w_0$. However, such an approach fails because after localization the noise rate would be possibly larger than the length of the Chow-Parameter that we want to estimate. This makes it impossible for us to learn the useful signal. On the other hand, [DKS18] gave a randomized localization method that can make the expected noise level sufficiently smaller than the length of the Chow-Parameter we want to estimate; and thus will succeed with constant probability in each round of refinement. Unfortinately, such an approach cannot be used in a query-efficient manner, because to implement such a method we need to know $\theta(w_i, w^*)$ up to an error $1/\log(1/\epsilon)$, in each round of refinement. This implies that if we make a random guess of $\theta(w_i, w^*)$, the probability of success in each round drops to only $1/\log(1/\epsilon)$, which requires to rerun the whole algorithm too many times in order to succeed once.

Such an issue could be addressed in a similar but more complicated way to the method we use in Lemma 3.5, by looking at the bias of the halfspace after localization. The second issue is that even the noise level is smaller than the length of the Chow-Parameter we want to estimate, the length of the Chow-Parameter is only $1/p^c$, for some small constant $c$, as we can only make $\theta_0$ smaller than some small constant. This still requires us to use $d/p^c$ queries to estimate it. Such an issue can again be addressed using the smoothed label method, where we use only $1/p^c$ queries to search a small class example and use another $\tilde{O}(d)$ queries to estimate the Chow-Parameter. Importantly, even such a method only succeeds with constant probability overall. As the refinement stage only runs for $O(\log\log(1/\epsilon))$ rounds, we only need to rerun the entire algorithm $O(\log(1/\epsilon))$ times to succeed once.

## Acknowledgement

Ilias Diakonikolas was supported by NSF Medium Award CCF-2107079, NSF Award CCF-1652862 (CAREER), a Sloan Research Fellowship, and a DARPA Learning with Less Labels (LwLL) grant. Daniel M. Kane was supported by NSF Medium Award CCF-2107547 and NSF Award CCF-1553288 (CAREER). Mingchen Ma was supported by NSF Award CCF-2144298 (CAREER).

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

## Supplementary Material

Here we give an organization of the supplementary material. In Appendix A, we present the formal definition of agnostic learning with membership queries and label queries. In Appendix B, we present the omitted proofs in Section 2 about the information-theoretical lower bound. In Appendix C, we discuss why we can without loss of generality assume the noise level $\mathrm{opt} \leq \epsilon$ and how to learn $p$ up to a constant factor with $\tilde{O}(1/p)$ queries. In Appendix D, we present the omitted proofs in Section 3.1 about how to learn a good hypothesis given a good initialization. In Appendix E.2, we present the omitted proofs in Section 3.2 about how to find a good initialization when the threshold is not extremely large. In Appendix F, we design an algorithm that finds a good initialization when the threshold is very large. In Appendix G, we prove Theorem 1.2.

## A  Active Learning with Membership Queries and Label Queries

**Definition A.1** (Active Learning Halfspace with Membership Queries)**.** *Let $H = \{h(x) = \mathrm{sign}(w \cdot x + t) : \mathbb{R}^d \to \{\pm 1\} \mid w \in S^{d-1}, t \geq 0\}$ be the class of halfspaces over $X = \mathbb{R}^d$. The labeling function $y(x) : X \to \{\pm 1\}$ is a random function that maps each $x \in X$ to an unknown binary random variable. For each $h \in H$, denote by $\mathrm{err}(h) = \mathbf{Pr}_{x \sim N(0,I)}(h(x) \neq y(x))$, $\mathrm{opt} := \min_{h \in H} \mathrm{err}(h)$ and $h^*(x) = \mathrm{sign}(w^* \cdot x + t^*)$ any halfspace with error $\mathrm{opt}$. A membership query takes $x \in X$ as an input and returns a label $y \sim y(x)$. We say a learning algorithm $\mathcal{A}$ is a constant-approximate learning algorithm if for every labeling function $y(x)$, and for every $\epsilon, \delta \in (0,1)$, it outputs some $\hat{h} \in H$ by adaptively making memberships queries, such that with probability at least $1 - \delta$, $\mathrm{err}(\hat{h}) \leq O(\mathrm{opt} + \epsilon)$. The query complexity of $\mathcal{A}$ is the total number of membership queries it uses during the learning process.*

**Definition A.2** (Active Learning Halfspace with Label Queries)**.** *Let $H = \{h(x) = \mathrm{sign}(w \cdot x + t) : \mathbb{R}^d \to \{\pm 1\} \mid w \in S^{d-1}, t \geq 0\}$ be the class of halfspaces over $X = \mathbb{R}^d$. Let $D$ be a distribution over $\mathbb{R}^d \times \{\pm 1\}$ such that $D_x$, the marginal distribution over $x$ is the standard Gaussian distribution $N(0,I)$. For each $h \in H$, denote by $\mathrm{err}(h) = \mathbf{Pr}_{(x,y) \sim N(0,I)}(h(x) \neq y)$, $\mathrm{opt} := \min_{h \in H} \mathrm{err}(h)$ and $h^*(x) = \mathrm{sign}(w^* \cdot x + t^*)$ any halfspace with error $\mathrm{opt}$. Let $S$ be a set of $m$ i.i.d. labeled examples drawn from $D$. An active learning algorithm with label query is given $S$ but with hidden labels and is allowed to make a label query for each $x \in S$ and see its label $y$. We say a learning algorithm $\mathcal{A}$ is a constant-approximate learning algorithm if for every distribution $D$ and for every $\epsilon, \delta \in (0,1)$, it outputs some $\hat{h} \in H$ by adaptively making label queries over a set of $m$ examples drawn i.i.d. from $D$, such that with probability at least $1 - \delta$, $\mathrm{err}(\hat{h}) \leq O(\mathrm{opt} + \epsilon)$. The label complexity of $\mathcal{A}$ is the total number label queries made over $S$ during the learning process*

## B  Omitted Proofs in Section 2

### B.1  Proof of Lemma 2.1

*Proof of Lemma 2.1.* Let $\mathcal{A}$ be such a learning algorithm. We select a random set of $m$ examples $S_1$ and give it to $\mathcal{A}$. With probability at least $2/3$, $\mathcal{A}$ makes $r$ queries and learns a halfspace $\hat{h}$ with error $p/2$ with respect to $h^*$. This implies that given a Gaussian example, with probability at least $p/2$ it will predict negative, and given it predicts negative, with probability at least $1/2$ it is actually negative. Since $m$ is at least $\mathrm{poly}(d, 1/p)$, we know that with enough high probability, at least $\Omega(d)$ examples will be predicted by negative by $\hat{h}$ and at least a constant fraction of these examples are actually negative. Thus, given such a $\hat{h}$ with probability at least $3/4$, we can find $d$ negative examples in $S$ by randomly querying $O(d)$ examples that are predicted as negative by $\hat{h}$. $\qquad\square$

### B.2  Proof of Lemma 2.2

Let $v = Aw^* = (w^* \cdot x_1, \ldots, w^* \cdot x_k)^\top$. Consider the projection of $w^*$ over the subspace spanned by the row vectors of $A$, $A^\top(AA^\top)^{-1}Aw^*$. Assuming that $x_1, \ldots, x_k$ are all negative, then $\|v\|^2 \geq$

$k(t^*)^2$. This implies that the square of the norm of the projection of $w^*$ onto the subspace is

$$B := (w^*)^\top A^\top (AA^\top)^{-1} Aw^* = v^\top (AA^\top)^{-1} v \geq \|v\|^2 / \left\|AA^\top\right\|_2 \geq k(t^*)^2 / \left\|AA^\top\right\|_2.$$

Since $w^*$ is uniformly chosen from the unit sphere, by Lemma B.1 in [KMT24], the square norm of $w^*$ projected onto a fixed $k-$dimensional subspace is a random variable drawn from a beta distribution $B(\frac{k}{2}, \frac{d-k}{2})$. By Lemma 2.2 in [DG03], if $\left\|AA^\top\right\|_2 \leq d(1 + O(1/(t^*)^2))$.

$$\mathbf{Pr}\left(B \geq k(t^*)^2 / \left\|AA^\top\right\|_2^2\right) \leq \exp\left(-\frac{k}{2}(\frac{d(t^*)^2}{\|AA^\top\|_2} - 1 - \log(\frac{d(t^*)^2}{\|AA^\top\|_2}))\right)$$

$$= \left(\sqrt{\frac{(t^*)^2 d}{\|AA^\top\|_2}} \exp(-\frac{1}{2}(\frac{d(t^*)^2}{\|AA^\top\|_2} - 1))\right)^k$$

$$\leq \left(O((t^*) \exp(-\frac{(t^*)^2}{2}(1 - O(1/(t^*)^2))))\right)^k$$

$$= \left(O((t^*)^2 \frac{1}{t^*} \exp(-\frac{(t^*)^2}{2}))\right)^k \leq (O(p\log(1/p)))^k.$$

The last inequality follows by Fact D.4.

### B.3 Proof of Lemma 2.3

We will first show that for a given $A \in \mathbb{R}^{k \times d}$, $\left\|AA^\top - dI\right\|_2$ is small with high probability if the rows of $A$ are drawn i.i.d. from $d$-dimensional standard Gaussian. Let $\mathcal{N}$ be an $1/4$-net of $S^{k-1}$. According to [Ver18], we know that $|\mathcal{N}| \leq e^{3k}$ and $\left\|AA^\top - dI\right\|_2 \leq 2\sup_{u \in \mathcal{N}}\left|u^t(AA^\top - dI)u\right|$. Thus, to show that $\left\|AA^\top - dI\right\|_2$ is small with high probability, it is equivalent to show with high probability for every $u \in \mathcal{N}$, $\left|u^\top(AA^\top - dI)u\right|$ is small.

Fix $u \in \mathcal{N}$ to be a unit vector. We have

$$u^\top AA^\top u - du^\top u = \sum_{j=1}^{d}(u^\top A_j)^2 - d.$$

Notice that each $u^\top A_j$ is a standard Gaussian variable and thus $\sum_{j=1}^{d}(u^\top A_j)^2$ is a chi-squared distribution with freedom $d$. By [Ver18], we have

$$\mathbf{Pr}\left(\left|\sum_{j=1}^{d}(u^\top A_j)^2 - d\right| \geq 2\xi d\right) \leq 2\exp\left(-d\xi^2/2\right).$$

Since $|\mathcal{N}| \leq e^{3k}$, we know that

$$\mathbf{Pr}\left(\left\|AA^\top - dI\right\|_2 \geq \xi d\right) \leq \mathbf{Pr}\left(\sup_{u \in \mathcal{N}}\left|u^\top(AA^\top - d)u\right| \geq 2\xi d\right) \leq 2\exp(3k - d\xi^2/2).$$

Since there are at most $\binom{m}{k}$ such $k$-tuples of examples, the probability that there exists a $k$-tuple such that $\left\|AA^\top - dI\right\|_2$ is larger than $\xi d = O(d/(t^*)^2)$ is at most

$$2\binom{m}{k}\exp(3k - d\xi^2/2) \leq 2\left(\frac{em}{k}\right)^k \exp(3k - d\xi^2/2) \leq 2\exp(-(d\xi^2/2 - 3k - k\log(er/k))) \leq 2/3,$$

by choosing $k = d/(\log(r)(t^*)^4)$ and $\xi = O(1/(t^*)^2)$.

## C   Omitted Details in Section 3

### C.1   Discussion on the Noise Level opt

Assuming we know some $\alpha$ such that $\epsilon \leq \alpha/2 \leq \mathrm{opt} \leq \alpha$, then learning $\hat{h}$ upto error $O(\mathrm{opt} + \epsilon)$ is equivalent to learning it up to error $O(\alpha)$. By guessing $\alpha = \epsilon 2^i$ for $i = 0, \ldots, O(\log(1/\epsilon))$, we

can always obtain a desired $\alpha$ and use it to run the learning algorithm and get a good hypothesis. Such an approach will generate a list of $O(\log(1/\epsilon))$ different hypotheses, finding a good enough hypothesis among them only costs $\mathrm{polylog}(1/\epsilon)$ queries using a standard tournament approach, such as the following lemma.

**Lemma C.1** (Lemma 3.6 in [DKK⁺24]). *Let $\epsilon, \delta \in (0,1)$ and $D$ a distribution over $\mathbb{R}^d \times \{0,1\}$. Given a list of hypothesises $\{h^{(i)}\}_{i=1}^k$, there is an algorithm that draws $O(k^2 \log(k/\delta)/\epsilon)$ unlabeled examples from $D_x$ and performs $O(k^2 \log(d/\delta))$ label queries runs in $\mathrm{poly}(d, \epsilon, \delta)$ times and output a hypothesis $\hat{h}$ such that*

$$\Pr_{(x,y)\sim D}(\hat{h}(x) \neq y) \leq 10 \min_{i\in[k]} \Pr_{(x,y)\sim D}(h^{(i)}(x) \neq y) + \epsilon.$$

### C.2 Approximate Bias Estimation Using Queries

In this part, we describe a simple approach to estimate the bias $p$ up to a constant factor using $\tilde{O}(1/p)$ queries. To do this we will estimate $\bar{p} = \Pr_{x\sim N(0,I)}(y(x) = -1)$, the noise version of $p$ as $|\bar{p} - p| \leq \epsilon$. If we can estimate $\hat{p}$ such that $\hat{p}/2 \leq \bar{p} \leq \hat{p}$ or verify that $\bar{p} \leq (C-1)\epsilon$, then $\hat{p}$ satisfies our purpose.

By Chebyshev's inequality, if $\bar{p} \leq 3\hat{p}/4$, then taking $O(1/\hat{p})$ random queries at $x$ and computing the empirical probability of $y(x) = -1$, with probability $2/3$, we are able to verify this fact by checking whether the empirical probability is less than $5\hat{p}/6$. On the other hand, if $4\hat{p}/5 \leq \bar{p} \leq \hat{p}$, with probability $2/3$ we are able to verify this fact by checking whether the empirical probability is greater than $5\hat{p}/6$. Furthermore, by repeating this approach $O(\log(1/\delta))$ times and using a majority voting trick, we can boost the probability of success up to $1 - \delta$. We will run the above approach for $\hat{p} = (4/5)^i/2$ for $i = 0, 1, \ldots$ until we find $\bar{p} \geq (4/5)\hat{p}$ or $\hat{p} = C'\epsilon$ for some constant $C'$. In the first case $(4/5)\hat{p} \leq \bar{p} \leq (25/24)\hat{p}$ and we find a good approximation for $\bar{p}$ and thus for $p$. In the second case, we can conclude that $p$ is smaller than $O(\epsilon)$

## D Omitted Proofs from Section 3.1

---

**Algorithm 3** REFINE(Learn the correct direction $w^*$ based on a warm start $w_0$)

---

**Input:** Intial direction $w_0 \in S^{d-1}$, $t' > 0$, an approximate threshold, error parameter $\epsilon \in (0,1)$, confidence parameter $\delta \in (0,1)$
**Output:** $w_T \in S^{d-1}$, an approximation of $w^*$, $\hat{t} \in \mathbb{R}$, an approximation of $t^*$
Let $\epsilon' = C\epsilon \exp(t'^2/2)$ $m = \tilde{O}(d)$, $T = O(\log(1/\epsilon'))$ $\sigma_0 \leftarrow \min\{1/t', 1/2\}$.
Let $C_1, C_2$ be large enough constants
**for** $i = 0, \ldots, T$ **do**
    $A_i \leftarrow I - (1 - \sigma_i^2)w_i w_i^t$, $\mu_i \leftarrow \sigma_i/C_1$
    Find $\tilde{t} \in \{0, \epsilon, 2\epsilon, \ldots, t'\}$ using the following binary search method, if no such $\tilde{t}$ is found, then stop the algorithm and **return** $w_T = 0$. ▷ **Find the correct threshold to construct the gradient.**
    Draw $O(\log(1/\delta))$ Gaussian samples $z \sim N(0,I)$, query $A_i^{1/2}z - \tilde{t}w_i$ and compute $p(\tilde{t})$, the empirical probability that a query returns $-1$. If $p(\tilde{t}) < e^{-17}$, properly decrease $\tilde{t}$, if $p(\tilde{t}) > e^{-17}$, properly increase $\tilde{t}$. Otherwise, declare that $\tilde{t}$ is found.
    **for** $j = 1, \ldots, m$ **do**
        Draw $z_j \sim N(0,I)$, make queries at $\tilde{z}_j := A_i^{1/2}z_j - \tilde{t}w_i$ and denote by $f_i(z_j)$ the result
    $\hat{g}_i \leftarrow \frac{1}{m}\sum_{j=1}^m \mathrm{proj}_{w_i^\perp}(z_j f_i(z_j))$          ▷ **Construct the gradient**
    $w_{i+1} \leftarrow \mathrm{proj}_{S^{d-1}}(w_i + \mu_i \hat{g}_i)$, $\sigma_{i+1} \leftarrow (1 - 1/C_2)\sigma_i$      ▷ **Gradient Descent**
$\hat{t} \leftarrow \tilde{t}$          ▷ **Use the threshold found in the last round**
**return** $w_T, \hat{t}$

---

### D.1 Proof of Lemma 3.2

*Proof of Lemma 3.2.* We first observe that

$$\|w_{i+1} - w^*\|^2 = \|\mathrm{proj}_{S^{d-1}}(w_i + \mu_i \hat{g}_i) - \mathrm{proj}_{S^{d-1}}(w^*)\|^2 \leq \|w_i + \mu_i \hat{g}_i - w^*\|^2.$$

It remains to upper bound $\|w_i + \mu_i \hat{g}_i - w^*\|^2$. We have

$$
\begin{aligned}
\|w_i + \mu_i \hat{g}_i - w^*\|^2 &= \|w_i - w^*\|^2 + 2\mu_i \hat{g}_i \cdot (w_i - w^*) + \mu_i^2 \|\hat{g}_i\|^2 \\
&= \|w_i - w^*\|^2 - 2\mu_i \hat{g}_i \cdot w^* + \mu_i^2 \|\hat{g}_i\|^2 \\
&= \|w_i - w^*\|^2 - 2\mu_i g_i \cdot w^* + 2\mu_i (g_i - \hat{g}_i) \cdot w^* + \mu_i^2 \|\hat{g}_i\|^2 \\
&= \|w_i - w^*\|^2 - 2\mu_i g_i \cdot w^* + 2\mu_i (g_i - \hat{g}_i) \cdot b_i u + \mu_i^2 \|\hat{g}_i\|^2 \\
&\leq \|w_i - w^*\|^2 - 2\mu_i g_i \cdot w^* + 2\mu_i b_i \|g_i - \hat{g}_i\| + \mu_i^2 \|\hat{g}_i\|^2 . \\
&= \|w_i - w^*\|^2 - 2\mu_i b_i g_i \cdot u + 2\mu_i b_i \|g_i - \hat{g}_i\| + \mu_i^2 \|\hat{g}_i\|^2 .
\end{aligned}
$$

Here, in the second equality, we use the fact that $\hat{g}_i \perp w_i$ and in the fourth equality, we use the fact that $(g_i - \hat{g}_i) \cdot w^* = (g_i - \hat{g}_i) \cdot a_i w_i + (g_i - \hat{g}_i) \cdot b_i u = (g_i - \hat{g}_i) \cdot b_i u$.

Next, we assume that $\sin(\theta_i/2) \leq \sigma_i$ and show that we can carefully choose parameter $\mu_i, \sigma_{i+1}$ to make $\sin(\theta_{i+1}/2) \leq \sigma_{i+1}$. We consider two cases. In the first case, we assume $3\sigma_i/4 \sin(\theta_i/2) \leq \sigma_i$. Since $\|w_i - w^*\| = 2\sin\frac{\theta_i}{2}$, by Lemma 3.2, we have

$$
\begin{aligned}
(2\sin\frac{\theta_{i+1}}{2})^2 &\leq (2\sin\frac{\theta_i}{2})^2 - 5\mu_i b_i + 2\mu_i c_2 b_i + \mu_i^2 c_1^2 \\
&\leq 4\sigma_i^2 - 15\sigma_i^2 c_1/(2C_1) + 4c_2 \sigma_i^2/C_1 + c_1^2 \sigma_i^2/C_1^2 \\
&\leq 4\sigma_i^2 - 5\sigma_i^2 c_1/(2C_1) + \sigma_i^2 c_1/(10C_1) + c_1^2 \sigma_i^2/C_1^2 \\
&\leq 4(1 - 5c_1/(8C_1) + c_1/(80C_1^2) + c_1^2/(2C_1^2))\sigma_i^2 := 4(1 - 1/C_2)^2 \sigma_i^2,
\end{aligned}
$$

where use the fact that $b_i \leq 2\sin(\theta_i/2) \leq 3\sigma_i/2$ and the fact that $C_1$ can be made large enough.

In the second case, we assume $\sin(\theta_i/2) < 3\sigma_i/4$. In this case, using the fact that

$$
2(\sin(\frac{\theta_{i+1}}{2}) - \sin(\frac{\theta_i}{2})) = \|w_{i+1} - w^*\| - \|w_i - w^*\| \leq \|w_{i+1} - w_i\| \leq \|w_i + \mu_i \hat{g}_i - w_i\| = \mu_i \|\hat{g}_i\| .
$$

We have

$$
\begin{aligned}
\sigma_{i+1} - \sin(\frac{\theta_{i+1}}{2}) &= \sigma_{i+1} - \sin(\frac{\theta_i}{2}) - (\sin(\frac{\theta_{i+1}}{2}) - \sin(\frac{\theta_i}{2})) \\
&\geq \sigma_{i+1} - \frac{3\sigma_i}{4} - \frac{\sigma_i \|\hat{g}_i\|}{C_1} \geq (\frac{1}{4} - \frac{1}{C_2} - \frac{1}{C_1})\sigma_i > 0,
\end{aligned}
$$

where the last inequality holds because the parameter $C_1, C_2$ can be chosen larger than 8.

$\square$

Lemma 3.2 implies that if $g_i$ has enough correlation with respect to $w_*$ but is also not too long, then by estimating $g_i$ up to some error, we can ensure $\|w_i - w^*\|$ drops significantly each round. Formally, we have the following corollary.

**Corollary D.1.** *In Algorithm 3, denote by $\theta_i = \theta(w^*, w_i)$. Assume that $\sin\frac{\theta_i}{2} \leq \sigma_i$. If there exist a suitable constant $c_1$ and a small enough constant $c_2$ such that $g_i \cdot w^* \geq c_1 \sigma_i/10$, $\|\hat{g}_i\| \leq c_1$ and $\|g_i - \hat{g}_i\| \leq c_2$. Then there exists large enough constant $C_1, C_2$ such that by taking $\mu_i = \sigma_i/C_1$, it holds that $\sin\frac{\theta_{i+1}}{2} \leq (1 - 1/C_2)\sigma_i$.*

*Proof of Corollary D.1.* Since $\|w_i - w^*\| = 2\sin\frac{\theta_i}{2}$, by Lemma 3.2, we have

$$
\begin{aligned}
(2\sin\frac{\theta_{i+1}}{2})^2 &\leq (2\sin\frac{\theta_i}{2})^2 - 5\mu_i \sigma_i + 2\mu_i c_2 \sigma_i + \mu_i^2 c_1^2 \\
&\leq 4\sigma_i^2 - 5\sigma_i^2 c_1/C_1 + 2c_2 \sigma^2/C_1 + c_1^2 \sigma_i^2/C_1^2 \\
&\leq 4\sigma_i^2 - 5\sigma_i^2 c_1/C_1 + 2\sigma^2/C_1^2 + c_1^2 \sigma_i^2/C_1^2 \\
&\leq 4(1 - 5c_1/(4C_1) + 1/(2C_1^2) + c_1^2/(2C_1^2))\sigma_i^2 := 4(1 - 1/C_2)^2 \sigma_i^2,
\end{aligned}
$$

where the third and the fourth inequalities hold when $C_1$ is large enough. $\square$

## D.2 Proof of Lemma 3.5

*Proof of Lemma 3.5.* We first prove Item 1. Since $T_i$ is a monotone decreasing function on $\tilde{t}$, and $t' > t^*$, it remains to show that for every $\tilde{t}$ such that $\left|\tilde{t} - t^*\right| \leq \sigma_i, |T_i| \leq 5$. Notice that

$$|T_i| = \left|\frac{t^* - a_i\tilde{t}}{\sigma_i\sqrt{a_i^2 + b_i^2/\sigma_i^2}}\right| \leq \left|\frac{t^* - a_i\tilde{t}}{\sigma_i}\right| \leq \left|\frac{\tilde{t} - a_i\tilde{t}}{\sigma_i}\right| + \left|\frac{\tilde{t} - t*}{\sigma_i}\right| \leq \frac{b_i^2\tilde{t}}{\sigma_i} + 1 \leq 5. \qquad (1)$$

By Fact 3.3, we know that

$$\bar{g}_i = \sqrt{\frac{2}{\pi}} \exp(-T_i^2/2)\frac{b_iu_i/\sigma_i}{\sqrt{a_i^2 + b_i^2/\sigma_i^2}}.$$

Since $\sqrt{a_i^2 + b_i^2/\sigma_i^2} \leq \sqrt{5}$ and $|T_i| \leq 5$, we immediately obatin Item 2. Finally, we prove Item 3. Using the monotone property of $T_i$, we prove the case where $\tilde{t} < t^* - 40\sigma_i$ and the case $\tilde{t} > t^* + 40\sigma_i$ can proved symmetrically. We have

$$T_i = \frac{t^* - a_i\tilde{t}}{\sigma_i\sqrt{a_i^2 + b_i^2/\sigma_i^2}} = \frac{t^* - \tilde{t}}{\sigma_i\sqrt{a_i^2 + b_i^2/\sigma_i^2}} + \frac{\tilde{t} - a_i\tilde{t}}{\sigma_i\sqrt{a_i^2 + b_i^2/\sigma_i^2}} \geq \frac{40\sigma_i}{\sqrt{5}\sigma_i} - 4 \geq 10,$$

where the first inequality holds because of Equation (1). $\qquad \square$

## D.3 Proof of Lemma 3.6

To prove Lemma 3.6, we first introduce the following definition called $(v, s, \sigma)$- rejection procedure.

**Definition D.2** ($(v, s, \sigma)$- rejection procedure). *Let $v \in \mathbb{R}^d$ be a unit vector and $s, \sigma$ be real numbers such that $\sigma < 1$. Given a point $x \in \mathbb{R}^d$, $(v, s, \sigma)$- rejection procedure accepts it with probability*

$$\exp\left(-(\sigma^{-2} - 1)(v \cdot x + s/(1 - \sigma^2))^2/2\right)$$

*and rejects it otherwise.*

$(v, s, \sigma)$- rejection procedure satisfies the following property.

**Lemma D.3** (Lemma C.7, Lemma C.8 in [DKS18]). *If $x \sim N(0, I)$ is fed into the $(v, s, \sigma)$- rejection procedure, then it is accepted with probability $\sigma \exp(-s^2/(2(1 - \sigma^2)))$. In particular, when $\sigma s \leq 2$ and $\sigma \leq 1/2$, the accepted probability is at least $\sigma \exp(-s^2/2 - 4)$. Moreover, the distribution on $x$ conditioned on acceptance is that of $N(-sv, A_{v,\sigma})$, where $A_{v,\sigma} = I - (1 - \sigma^2)vv^t$.*

*Proof of Lemma 3.6.* Let $\tilde{z} := A_i^{1/2}z - \tilde{t}w_i$. By Fact 3.3, we know that $\ell_i(z) = h^*(\tilde{z}), \forall z \in \mathbb{R}^d$. By Lemma 3.6, we know that if $z \sim N(0, I)$, then $\tilde{z} \sim N(-\tilde{t}w_i, A_i)$, which can be seen by feeding a Gaussian random vector into the $(w_i, \tilde{t}, \sigma_i)-$rejection procedure conditioned on acceptance. Since $\mathrm{err}(h^*) = \mathrm{opt} \leq \epsilon$ and the accepted rate is at least $\sigma \exp(-s^2/2 - 4)$, we know from Lemma 3.6 that

$$\Pr_{z \sim N(0,I)} (\ell_i(z) \neq f_i(z)) = \Pr_{z \sim N(0,I)} (h^*(\tilde{z}) \neq f_i(z)) \leq \epsilon \exp(\tilde{t}^2/2 + 4)/\sigma_i.$$

In particular, if $\sigma_i \geq C\exp((t')^2/2)\epsilon$, we have

$$\epsilon\exp(\tilde{t}^2/2 + 4)/\sigma_i \leq \left(\epsilon\exp(\tilde{t}^2/2 + 4)\right)/\left(C\epsilon\exp((t')^2/2)\right) \leq e^4/C := c \leq e^{-40}.$$

$\qquad \square$

## D.4 Proof of Lemma 3.7

Before presenting the proof, we state the following two facts that will be used in our proof.

**Fact D.4** (Komatsu's Inequality). *For any $t \in \mathbb{R}$ the bias $p$ of a halfspace $h(x) = \mathrm{sign}(w^* \cdot x + t)$ can be bounded as*

$$\sqrt{\frac{2}{\pi}}\frac{\exp(-t^2/2)}{t + \sqrt{t^2 + 4}} \leq p \leq \sqrt{\frac{2}{\pi}}\frac{\exp(-t^2/2)}{t + \sqrt{t^2 + 2}}.$$

**Fact D.5** (Lemma B.4 in [DKTZ22]). *Let $D$ be a distribution on $\mathbb{R}^d \times \{\pm 1\}$ with standard normal $x-$margin and let $w, u$ be two orthogonal unit vectors. Let $B$ be any interval over $\mathbb{R}$ and let $S(x, y)$ be any event over $\mathbb{R}^d \times \{\pm 1\}$, such that $S(x, y) \subseteq \{w \cdot x \in B\}$ then it holds*

$$\mathop{\mathbf{E}}_{D} (|u \cdot x| \mathbf{1}\{S(x, y)\}) \leq 2\sqrt{e} \, \mathbf{Pr}(S(x, y)) \sqrt{\log(\frac{\mathbf{Pr}(w \cdot x \in B)}{\mathbf{Pr}(S(x, y))})}.$$

*Proof of Lemma 3.7.* We start by proving the first part of Lemma 3.7. By Lemma 3.6, we know that $\eta_i := \mathbf{Pr}_{z \sim N(0,I)}(\ell_i(z) \neq f_i(z)) \leq e^{-40}$. This implies that $\left|\mathbf{Pr}_{z \sim N(0,I)}(\ell_i(z) = -1) - p_i\right| \leq e^{-40}$. We first show that when $p_i$ is in a reasonable range, $|T_i| < 6$. Assuming by contradiction that $|T_i| \geq 6$, then by Fact D.4, the bias of $\ell_i(z)$ must be at most $\exp(-T_i^2/2)/(2T_i) \leq e^{-20}$, which implies that it cannot be the case where $p_i \in (e^{-18}, 1 - e^{-18})$. Similarly, if $|T_i| < 5$, then by Fact D.4, the bias of $\ell_i(z)$ must be at least $\exp(-T_i^2/2)/20 \geq e^{-15.5}$. As the noise level $\eta_i \leq e^{-40}$, we have $p_i \in (e^{-18}, 1 - e^{-18})$.

Next, we prove the second part of Lemma 3.7. We start by bounding the correlation between $g_i$ and $u_i$. We have

$$g_i \cdot u_i = \mathop{\mathbf{E}}_{z \in N(0,I)} \text{proj}_{w_i^\perp} z(\ell_i(z) + f_i(z) - \ell_i(z)) \cdot u_i$$

$$= \bar{g}_i \cdot u_i - \mathop{\mathbf{E}}_{z \in N(0,I)} \text{proj}_{w_i^\perp} z(\ell_i(z) - f_i(z)) \cdot u_i$$

$$\geq \bar{g}_i \cdot u_i - \mathop{\mathbf{E}}_{z \in N(0,I)} |u_i \cdot z| \mathbf{1}\{\ell_i(z) \neq f_i(z)\}$$

$$\geq \bar{g}_i \cdot u_i - 2\sqrt{e} \eta_i \sqrt{\log(1/\eta_i)},$$

where the third and the last inequalities hold because $u_i \perp w_i$ and Fact D.5.

We next bound the norm of $g_i$. Since both $\bar{g}_i$ and $g_i$ are orthogonal to $w_i$. It is sufficient to show that for every unit vector $u \perp w_i$, $|g_i \cdot u| \leq |\bar{g}_i \cdot u| + 4\sqrt{e} \eta_i \sqrt{\log(1/\eta_i)}$. We have

$$|g_i \cdot u| = \left| \mathop{\mathbf{E}}_{z \in N(0,I)} \text{proj}_{w_i^\perp} z(\ell_i(z) + f_i(z) - \ell_i(z)) \cdot u \right|$$

$$= \left| \bar{g}_i \cdot u_i - \mathop{\mathbf{E}}_{z \in N(0,I)} \text{proj}_{w_i^\perp} z(\ell_i(z) - f_i(z)) \cdot u_i \right|$$

$$\leq |\bar{g}_i \cdot u_i| + \mathop{\mathbf{E}}_{z \in N(0,I)} |u_i \cdot z| \mathbf{1}\{\ell_i(z) \neq f_i(z)\}$$

$$\leq |\bar{g}_i \cdot u_i| + 2\sqrt{e} \eta_i \sqrt{\log(1/\eta_i)}.$$

$\square$

## D.5 Proof of Theorem 3.1

In this section, we present the proof of Theorem 3.1. Before presenting the proof, we present the following fact that will be a crucial part of our proof.

**Fact D.6** (Lemma 4.2 in [DKS18]). *Under the standard normal distribution for every pair of unit vector $w, w^*$ and real number $t$,*

$$\mathbf{Pr}(\text{sign}(w^* \cdot x + t) \neq \text{sign}(w \cdot x + t)) \leq \frac{\sin(\theta(w, w^*))}{2} \exp(-t^2/2).$$

*Proof of Theorem 3.1.* Denote by $\theta_i := \theta(w_i, w^*)$. We will first show by induction that with high probability in the $i$-th round of Algorithm 3, $\sin(\theta_i/2) \leq \sigma_i$. Assuming this is correct, since $\sigma_T = C\epsilon \exp(t')^2/2$ for some large constant $C$. We will have

$$\mathbf{Pr}(\text{sign}(w^* \cdot x + t^*) \neq \text{sign}(w \cdot x + t^*)) \leq C\epsilon \exp(\frac{(t' + t^*)(t' - t^*)}{2}) \leq C\epsilon \exp(\frac{1}{\sqrt{\log(1/\epsilon)}}) = O(\epsilon). \tag{2}$$

Since opt $\leq \epsilon$, this implies that by providing a good enough estimation of $t^*$, we found a hypothesis with error at most $O(\epsilon)$. Now we show that this is actually true. For $i = 0$, $\sin(\theta_0/2) \leq \sigma_0$ holds by our assumption.

Now, we assume this is correct for the $i$-round and we show this holds with high probability for the $i+1$-th round. We will show that with high probability the gradient $\hat{g}_i$ we use in the update $w_{i+1} = \text{proj}_{w_i^\perp}(w_i + \mu_i \hat{g}_i)$ satisfies the condition of Lemma 3.2.

Recall that we have the following notations. $w^* = a_i w_i + b_i u_i$. $T_i := \frac{t^* - a_i \tilde{t}}{\sigma_i \sqrt{a_i^2 + b_i^2/\sigma_i^2}}$, $\ell_i(z) = \text{sign}((a_i w_i + b_i u_i/\sigma_i)z + (t^* - a\tilde{t})/\sigma_i)$, $\bar{g}_i = \mathbf{E}_{z \in N(0,I)} \text{proj}_{w_i^\perp} z \ell_i(z)$ and $g_i = \mathbf{E}_{z \in N(0,I)} \text{proj}_{w_i^\perp} z f_i(z)$, where $f_i(z) = y(A_i^{1/2} z - \tilde{t} w_i)$. And $\eta_i := \mathbf{Pr}_{z \sim N(0,I)}(\ell_i(z) \neq f_i(z)) < e^{-40}$ by Lemma 3.6.

We first show that with high probability, Algorithm 3 must be able to select a correct threshold $\tilde{t} \leq t'$ such that $|T_i| \leq 6$. Denote by $p_i$ the probability that $f_i(z) = -1$. We notice that for each fixed $\tilde{t}$ by randomly querying $O(\log(1/\delta))$ $f_i(z)$, we can with high probability check if $p_i \in (e^{-17}, 1 - e^{-17})$ or not. This can be done using the same method we used in Appendix C.2.

Since $b_i/2 = \sin\theta_i/2 \leq \sin(\theta_i/2) \leq \sigma_i$, we know from Lemma 3.7 that as long as we find some $\tilde{t}$ such that $p_i \in (e^{-17}, 1 - e^{-17})$, we have have $|T_i| \leq 6$. By Lemma 3.5 and Lemma 3.7, we know that there exists an interval $I_i \subseteq [0, t']$ of length at least $\sigma_i > \epsilon$ such that for every $\tilde{t} \in I_i$, $|T_i| < 5$ and thus $p_i \in (e^{-16}, 1 - e^{-16})$. Thus, by performing a binary search at most $O(\log(1/\epsilon))$ times, with high probability, we are able to find such a $\tilde{t}$ such that $|T_i| < 6$. Given that we find such a correct $\tilde{t}$, we will consider two cases.

First, we assume that $3\sigma_i/4 \leq \sin(\theta_i/2) \leq \sigma_i$. We will show that with high probability $g_i$ and its empirical estimation $\hat{g}_i$ satisfy the condition in the statement of Corollary D.1 and thus prove $\sin(\theta_{i+1}/2) \leq \sigma_{i+1}$. Since $\text{proj}_{w_i^\perp} z f_i(z)$ is 1-subgaussian random vector, by Hoeffding's inequality, we know that with $\tilde{O}(d)$ samples of $z$, with high probability we have $\|g_i - \hat{g}_i\| \leq c_2 \leq e^{-40}$.

By Lemma 3.7 and Lemma 3.5, we have

$$g_i \cdot u_i = \bar{g}_i \cdot u_i - 2\sqrt{e}\eta_i\sqrt{\log(1/\eta_i)} \geq \frac{b_i \|\bar{g}_i\|}{\sigma_i} e^{-19} - 100e^{-40} \geq e^{-19} - 100e^{-40} \geq e^{-20} := c_1.$$

$$\|\hat{g}_i\| \leq \|g_i\| + \|g_i - \hat{g}_i\| \leq \|\bar{g}_i\| + 2\sqrt{e}\eta_i\sqrt{\log(1/\eta_i)}) + e^{-40} \leq 3e^{-19} \leq 10c_1.$$

Thus, by Corollary D.1, we can conclude that $\sin(\theta_{i+1}/2) \leq (1 - 1/C_2)\sigma_i = \sigma_{i+1}$, for a large constant $C_2$.

Next, we consider the case where $\sin(\theta_i/2) < 3\sigma_i/4$. In this case, as we have shown that $\|\hat{g}_i\|$ is bounded by some universal constant, the condition of Lemma 3.2 is fulfilled automatically and thus $\sin(\theta_{i+1}/2) \leq (1 - 1/C_2)\sigma_i = \sigma_{i+1}$, for a large constant $C_2$.

By induction, with a high probability for each $i$, we have $\sin(\theta_i/2) \leq \sigma_i$ and thus $w_T$ is a good approximation of $w^*$. It remains to show that $\hat{t}$ is also a good approximation of $t^*$. Recall that $\hat{t} = \tilde{t} < t'$ such that $|T_T| < 6$. Lemma 3.5 implies that $|\hat{t} - t^*| \leq 40\sigma_T = 40C\epsilon \exp(t')^2/2$. Thus,

$$\mathbf{Pr}_{x \sim N(0,I)} \left(\text{sign}(w_T \cdot x + t^*) \neq \text{sign}(w_T \cdot x + \hat{t})\right) \leq (2\pi)^{-1} |t^* - \hat{t}| \exp(-\frac{(t^* - |t^* - \hat{t}|)^2}{2})$$

$$\leq (2\pi)^{-1} 40C\epsilon \exp(\frac{(t')^2 - (t^* - |t^* - \hat{t}|)^2}{2})$$

$$\leq (2\pi)^{-1} 40C\epsilon \exp(2t'(t' - t^* + 40\sigma_T))$$

$$\leq (2\pi)^{-1} 40C\epsilon \exp(2t'(40\sigma_T + \frac{1}{\log(1/\epsilon)}))$$

$$= O(\epsilon \exp(80t'\sigma_T)).$$

Since $\sigma_T t' = O(\epsilon t' \exp((t')^2/2))$ and $t' \exp((t')^2/2) \leq 1/(C\epsilon)$, we can conclude that $\mathbf{Pr}_{x \sim N(0,I)} \left(\text{sign}(w_T \cdot x + t^*) \neq \text{sign}(w_T \cdot x + \hat{t})\right) \leq O(\epsilon)$. Thus, with high probability $\text{err}(\text{sign}(w_T \cdot x + \hat{t})) \leq O(\epsilon)$.

Finally, we count the number of queries used by Algorithm 3. In each round of the algorithm, we perform $O(\log(1/\epsilon))$ binary searches to find the correct parameter $\hat{t}$, each of which takes us only $\tilde{O}(1)$ queries. We also make $\tilde{O}(d)$ queries to construct $\hat{g}_i$ in each round of the algorithm. Thus, each round of Algorithm 3 takes $\tilde{O}(d + \log(1/\epsilon))$ queries. Since there are at most $O(\log(1/\epsilon))$ rounds, the query complexity of Algorithm 3 is $\tilde{O}(d\,\mathrm{polylog}(1/\epsilon))$.

$\qquad\square$

## E   Omitted Proofs from Section 3.2

### E.1   Proof of Lemma 3.11

*Proof of Lemma 3.11.* Denote by $D^-$ the conditional distribution of $x \sim N(0, I)$ given $y(x) = -1$. Recall that

$$\Pr_{x \sim N(0,I)}(y(x) = -1) \geq \Pr_{x \sim N(0,I)}(h^*(x) = -1) - \epsilon \geq (1 - 1/C)p.$$

We will first show that $E_{x \sim D^-}\eta(x) \leq O(\epsilon/p)$. Denote by $Z := \sqrt{1 - \rho^2}x + \rho z$, where $x \sim D^-, z \sim N(0, I)$, then

$$\mathop{\mathbf{E}}_{x \sim D^-} \eta(x) = \Pr_{x \sim D^-, z \sim N(0,I)} \mathbf{1}\{h^*(\sqrt{1 - \rho^2}x + \rho z) \neq y(\sqrt{1 - \rho^2}x + \rho z)\} = \Pr_Z \mathbf{1}\{h^*(Z) \neq y(Z)\}.$$

Since $z$ and $x$ are independent, we notice that $Z$ can be simulated via the following reject sampling process. We draw $x \sim N(0, I)$ and $Z \sim N(0, I)$ to construct $Z = \sqrt{1 - \rho^2}x + \rho z$ and accepted $Z$ when $y(x) = -1$. Since $\sqrt{1 - \rho^2}N(0, I) + \rho N(0, I) = N(0, I)$, $Z$ can be seen as a reject sampling process with an accepted rate at least $(1 - 1/C)p$. Since the noise rate $\mathrm{opt} \leq \epsilon$, we know that

$$\mathop{\mathbf{E}}_{x \sim D^-} \eta(x) = \Pr_Z \mathbf{1}\{h^*(Z) \neq y(Z)\} \leq (1 - 1/C)^{-1}\epsilon/p.$$

By Markov's inequality, we know that with probability at least $3/4$, $\eta(x) \leq 5\epsilon/p$, with $x \sim D^-$.

Next, we show that with a constant probability a negative example $x$ must be close to the decision boundary of $h^*$. We have

$$\Pr_{x \sim D^-}\left(w^* \cdot x < -t^* - \frac{1}{t^*}\right)$$

$$= \Pr_{x \sim D^-}(h^*(x) = -1)\Pr_{x \sim D^-|\{h^*(x)=-1\}}\left(w^* \cdot x < -t^* - \frac{1}{t^*}\right)$$

$$+ \Pr_{x \sim D^-}(h^*(x) = +1)\Pr_{x \sim D^-|\{h^*(x)=+1\}}\left(w^* \cdot x < -t^* - \frac{1}{t^*}\right)$$

$$\leq \Pr_{x \sim D^-|\{h^*(x)=-1\}}\left(w^* \cdot x < -t^* - \frac{1}{t^*}\right) + 1/C \leq \Pr_{x \sim N(0,I)|\{h^*(x)=-1\}}\left(w^* \cdot x < -t^* - \frac{1}{t^*}\right) + 2/C$$

$$= \int_{t^*+1/t^*}^{\infty} \exp(-s^2/2)ds / \int_{t^*}^{\infty} \exp(-s^2/2)ds + 2/C \leq \exp(-\frac{(t^* + \frac{1}{t^*})^2 - (t^*)^2}{2}) + 2/C$$

$$= \exp(-\frac{(2t^* + 1/t^*)/t^*}{2}) + 2/C \leq e^{-1} + 2/C,$$

where in the third inequality, we use Fact D.4.

Thus, by union bound, with probability at least $1/2$, it simultaneously holds that $\eta(x) \leq 5\epsilon/p$ and $w^* \cdot x \leq -t^* - 1/t^*$.

$\qquad\square$

## E.2   Proof of Theorem 3.8

*Proof of Theorem 3.8.* We consider two cases. First, if $t' < 1$, then each $x_0^{(i)} = z_i$ is drawn from the standard Gaussian. We have

$$
\begin{aligned}
\left\| u_0 - \mathop{\mathbf{E}}_{z \sim N(0,I)} z h^*(z) \right\| &= \left\| u_0 - \mathop{\mathbf{E}}_{z \sim N(0,I)} z y(z) + \mathop{\mathbf{E}}_{z \sim N(0,I)} z y(z) - \mathop{\mathbf{E}}_{z \sim N(0,I)} z h^*(z) \right\| \\
&\leq \left\| u_0 - \mathop{\mathbf{E}}_{z \sim N(0,I)} z y(z) \right\| + \left\| \mathop{\mathbf{E}}_{z \sim N(0,I)} z y(z) - \mathop{\mathbf{E}}_{z \sim N(0,I)} z h^*(z) \right\| \\
&\leq \left\| u_0 - \mathop{\mathbf{E}}_{z \sim N(0,I)} z y(z) \right\| + \sup_{u \in S^{d-1}} \mathop{\mathbf{E}}_{z \sim N(0,I)} |u \cdot z| \, \mathbf{1}(y(z) \neq h^*(z)) \\
&\leq \left\| u_0 - \mathop{\mathbf{E}}_{z \sim N(0,I)} z y(z) \right\| + 2\sqrt{e}\epsilon \sqrt{\log(1/\epsilon)},
\end{aligned}
$$

where the last inequality holds because of Fact D.5. Since each $z_i y(z_i)$ is a standard Gaussian, by Hoeffding's inequality, we have

$$
\mathbf{Pr}\left( \left\| u_0 - \mathop{\mathbf{E}}_{z \sim N(0,I)} z y(z) \right\| \geq r \leq 2 \exp\left(-\frac{mr^2}{d}\right) \leq \mathrm{polylog}(\delta),
$$

when $m \geq \tilde{\Omega}(d/r^2)$. By taking $r = (20 \log(1/\epsilon))^{-1}$, we obtain that $\left\| u_0 - \mathbf{E}_{z \sim N(0,I)} z y(z) \right\| \leq (20 \log(1/\epsilon))^{-1}$ with high probability. Thus

$$
\left\| u_0 - \mathop{\mathbf{E}}_{z \sim N(0,I)} z h^*(z) \right\| \leq (20 \log(1/\epsilon))^{-1} + 2\sqrt{e}\epsilon \sqrt{\log(1/\epsilon)} \leq O(\log(1/\epsilon)^{-1}).
$$

By Fact 3.4, we know that $\mathbf{E}_{z \sim N(0,I)} z h^*(z) = \xi w^*$ for some $\xi \geq e^{-1}$, which also implies that $\|u_0\| \geq e^{-1}/2$, because $u_0$ sufficiently close to $\mathbf{E}_{z \sim N(0,I)} z h^*(z)$.
Since

$$
\begin{aligned}
\left\| u_0 - \mathop{\mathbf{E}}_{z \sim N(0,I)} z h^*(z) \right\|^2 &= \|u_0\|^2 + \left\| \mathop{\mathbf{E}}_{z \sim N(0,I)} z h^*(z) \right\|^2 - 2 \left\| \mathop{\mathbf{E}}_{z \sim N(0,I)} z h^*(z) \right\| \|u_0\|^2 \cos\theta(w_0, w^*) \\
&\geq 2 \left\| \mathop{\mathbf{E}}_{z \sim N(0,I)} z h^*(z) \right\| \|u_0\| (1 - \cos\theta(w_0, w^*)) \\
&= 4 \left\| \mathop{\mathbf{E}}_{z \sim N(0,I)} z h^*(z) \right\| \|u_0\| \sin^2(\theta(w_0, w^*)/2),
\end{aligned}
$$

we get $\sin(\theta(w_0, w^*)/2) \leq \sqrt{\left\| u_0 - \mathbf{E}_{z \sim N(0,I)} z h^*(z) \right\|^2 / 4 \left\| \mathbf{E}_{z \sim N(0,I)} z h^*(z) \right\| \|u_0\|} \leq O(1/\log(1/\epsilon))$. In particular as $\epsilon < 1/C$ for some large enough $C$, we conclude that $\sin(\theta(w_0, w^*)/2) \leq \max\{\min\{1/t, 1/2\}, O(\eta \sqrt{\log(1/\eta)})\}$.

We next address the case when $t > 1$. By Lemma 3.11, we know that with probability at least $1/2$, we have $\eta(x) \leq 5\epsilon/p$ and $w^* \cdot x \in (-t^* - 1/t^*, -t^*)$. We will assume these two events happen in the rest of the proof. Let $z \sim N(0, I)$ and by Fact 3.10, define

$$
\tilde{h}(z) := h^*(\tilde{x}_0) = \mathrm{sign}(w^* \cdot z + \frac{t^* + \sqrt{1 - \rho^2} w^* \cdot x_0}{\rho}).
$$

By Lemma 3.11, we know that $\mathbf{Pr}_{z \sim N(0,I)} \tilde{h}(z) \neq \tilde{y}(x_0) = \eta(x_0) \leq 5\epsilon/p$. Similar to the first case, we have

$$\left\| u_0 - \underset{z \sim N(0,I)}{\mathbf{E}} z\tilde{h}(z) \right\| = \left\| u_0 - \underset{z \sim N(0,I)}{\mathbf{E}} z\tilde{y}(x_0) + \underset{z \sim N(0,I)}{\mathbf{E}} z\tilde{y}(x_0) - \underset{z \sim N(0,I)}{\mathbf{E}} z\tilde{h}(z) \right\|$$

$$\leq \left\| u_0 - \underset{z \sim N(0,I)}{\mathbf{E}} z\tilde{h}(z) \right\| + \left\| \underset{z \sim N(0,I)}{\mathbf{E}} z\tilde{h}(z) - \underset{z \sim N(0,I)}{\mathbf{E}} z\tilde{h}(z) \right\|$$

$$\leq \left\| u_0 - \underset{z \sim N(0,I)}{\mathbf{E}} z\tilde{y}(x_0) \right\| + \sup_{u \in S^{d-1}} \underset{z \sim N(0,I)}{\mathbf{E}} |u \cdot z| \, \mathbf{1}(\tilde{y}(x_0) \neq \tilde{h}(z))$$

$$\leq \left\| u_0 - \underset{z \sim N(0,I)}{\mathbf{E}} z\tilde{y}(x_0) \right\| + 2\sqrt{e}\eta(x_0)\sqrt{\log(1/\eta(x_0))}$$

$$\leq \max\{O(\eta(x_0)\sqrt{\log(1/\eta(x_0))}), 1/(50\sqrt{\log(1/\epsilon)})\}$$

$$\leq \max\{O(\eta\sqrt{\log(1/\eta)}), 1/(50t)\} \,,$$

where in the second last inequality we use the fact that $\left\| u_0 - \mathbf{E}_{z \sim N(0,I)} z\tilde{y}(x_0) \right\| \leq 1/(100\sqrt{\log(1/\epsilon)}) \leq 1/(100t)$ with high probability. Since $\rho = 1/t$, $|t - t^*| \leq 1/\log(1/\epsilon)$ and $t^* \leq \sqrt{\log(1/\epsilon)} \ll \log(1/\epsilon)$, the threshold $T_\rho = \frac{t^* + \sqrt{1-\rho^2} w^* \cdot x_0}{\rho}$ can be bounded as follows.

$$-1 \leq \frac{t^* - \sqrt{1-\rho^2}(t^* + \frac{1}{t^*})}{\rho} \leq T_\rho \leq \frac{t^*(1 - \sqrt{1-\rho^2})}{\rho} \leq tt^*/(t)^2 \leq 1 + o(1).$$

Fact 3.4 implies that $\mathbf{E}_{z \sim N(0,I)} z\tilde{h}(z) = \xi w^*$ for some $\xi \geq e^{-1}$. Since $u_0$ is close to $\mathbf{E}_{z \sim N(0,I)} z$, $\|u_0\| \geq e^{-1}/2$. Thus, we obtain that

$$\sin(\theta(w_0, w^*)/2) \leq \sqrt{\left\| u_0 - \underset{z \sim N(0,I)}{\mathbf{E}} z\tilde{h}(z) \right\|^2 / 4 \left\| \underset{z \sim N(0,I)}{\mathbf{E}} zh^*(z) \right\| \|u_0\|}$$

$$\leq \max\{\min\{1/t, 1/2\}, O(\eta\sqrt{\log(1/\eta)}\}.$$

$\square$

## F    Finding a Good Initialization with an Extreme Threshold

By Theorem 3.8, we know that Algorithm 2 can only find some $w_0$ such that $\sin(\theta_0/2) \leq O(\eta\sqrt{\log(1/\eta)})$, where $\eta = \epsilon/p$ when $p$ is small such that $\eta\sqrt{\log(1/\eta)} > O(1/t)$. In this section, we design an algorithm that finds a warm start with a non-negligible probability of success when the threshold $t^*$ falls in this range. Formally, we prove the following theorem.

**Theorem F.1.** *Let $h^*(x) = \text{sign}(w^* \cdot x + t^*)$ be a halfspace with bias $p$ and $y(x)$ be any labeling function such that $\text{err}(h^*) = \text{opt} \leq \epsilon \leq 1/C$ for some large enough constant $C$. Let $t$ be a scalar such that $t - 1/\log(1/\epsilon) \leq t^* \leq t$ and $1/(400t) \leq \eta\sqrt{\log(1/\eta)} \leq 1/C$ for some large enough constant $C$, where $\eta = \epsilon/p$, Algorithm 5 makes $M = \tilde{O}(1/p + d\log(1/\epsilon))$ membership queries, runs in $\text{poly}(d, M)$ time and with probability at least $1/\text{polylog}(1/\epsilon)$, outputs some $w_0$ such that $\sin(\theta(w_0, w^*)/2) \leq 1/t$.*

The high-level idea of our algorithm is as follows. Although Algorithm 2 will not provide us a $w_0$ such that $\theta_0 \leq O(1/t)$, $\theta_0$ is still smaller than a sufficiently small constant. We want to use the localization technique to refine $w_0$ so that after $T$ rounds of refinement, $\sin(\theta_T/2) \leq \sigma_T = 1/t$. Recall in Appendix D, we introduce Definition D.2, $(v, s, \sigma)$-rejection procedure, which can be simulated using membership query. Passing a Gaussian random point to the $(v, s, \sigma)$-rejection procedure, according to Lemma D.3, we will get a another distribution over $\mathbb{R}^d \times \{\pm 1\}$ that behaves the same as another halfspace $h'$.

In this section, we want to design a $(v, s, \sigma)$-rejection procedure such that the direction of the halfspace $h'$ has a constant correlation with respect to $w^*$ and the noise level after the rejection procedure is much smaller than the length of the Chow-Parameter of $h'$. Write $w^* = a_i w_i + b_i u_i$. We want to set up $v = w_i$, $\sigma = 1/t$ and $s \sim (a_i t, a_i t + b_i)$ uniformly. Such a method is called the randomized threshold method in [DKS18]. This method has the following property.

**Lemma F.2** (Proposition C.11 in [DKS18])**.** *Let $a, b, t > 0$ such that $a^2 + b^2 = 1$ and $t$ larger than some constant $C$. Let $w \in S^{d-1}$. Let $s \sim [at, at + b]$ uniformly. For each $x \in \mathbb{R}^d$, the expected probability that $x$ is accepted by the $(w, s, \sigma)$-rejection procedure is at most $\sigma/b$, where $\sigma = 1/t$.*

Lemma F.2 implies that in expectation over the randomness of $s$, only $\sigma/b_i$-fraction of the noisy points will pass the $(w_i, s, \sigma)$-rejection procedure. If we use query to simulate such a rejection procedure, by Lemma D.3, with a constant probability, the noise rate among our queries would be $O(\epsilon \exp(s^2/2)/b_i)$. However, as we do not know $b_i$, using some $b$ that is slightly far from $b_i$ would make the noise level too high for us to learn the signal we want. To overcome this, we design the following test approach to show that given a $b$, we can with high probability check if it can be used to construct the rejection procedure or not and in particular, when $b - 1/\log(1/\epsilon) < b_i < b$, such a $b$ is guaranteed to pass our test.

**Lemma F.3.** *Let $h^*(x) = \text{sign}(w^* \cdot x + t^*)$ be a halfspace and $y(x)$ be any labeling function such that $\text{err}(h^*) = \text{opt} \leq \epsilon$. Let $w \in S^{d-1}$ be unit vector such that $w^* = a^* w + b^* u$, $a^*, b^* > 0$ and $(a^*)^2 + (b^*)^2 = 1$, $\bar{b} < 1/4$. Let $t > 0$ such that $t \exp(t^2/2) \leq 1/(C\epsilon)$ for a sufficiently large constant $C$. Let $a, b \in (0, 1)$ such that $a^2 + b^2 = 1$. Let $b, t, w, \delta$ be input of Algorithm 4. Let $s \sim (at, at + b)$ uniformly. Denote by $p(b, s)$ be bias of a halfspace with threshold $T_{bs} := (t - as)/b$. Let $\ell(z) = \text{sign}((a^* \sigma w + b^* u) \cdot z + t^* - a^* s)$ be a halfspace with bias $p_s$, where If the probability that $p_s > p(b, s)/4$ is at most $1/2$, then with probability at least $1 - \delta$, Algorithm 4 output "No". If the probability that $p_s > p(b, s)/2$ is at least $29/30$, then with probability at least $1 - \delta$, Algorithm 4 output "Yes". Furthermore, the query complexity of Algorithm 4 is $\tilde{O}_\delta(1/p^2(b, at)) = \tilde{O}_\delta(1/\sqrt{p})$*

*In particular, when $b^* \geq 1.5/t \geq 1.5/\sqrt{\log(1/\epsilon)}$, $|b - b^*| \leq 1/\log(1/\epsilon)$ and $|t - t^*| \leq 1/\log(1/\epsilon)$, with probability at least $1 - \delta$, Algorithm 4 will output "Yes".*

---

**Algorithm 4** ANGLE TEST(Check if $b$ is a good approximation for $\sin\theta(w^*, w)$)

---

**Input:** A direction $w$, confidence parameter $\delta \in (0, 1)$, threshold $t > 0$, parameter $b$
**Output: "Yes" or "No"**
Count $\leftarrow 0$.
$A \leftarrow I - (1 - \sigma^2) w w^t$, $\sigma = 1/t$
Compute $a = \sqrt{1 - b^2}$
Let $T_{bs} = (t - as)/b$ and $p(b, s)$ be the bias of a halfspace with threshold $T_{bs}$
**for** $i = 1 \ldots T = O(\log(1/\delta))$ **do**
    Draw $s \sim [at, at + b]$ uniformly
    Draw $m = \tilde{O}(1/p^2(b, s))$ $z \sim N(0, I)$ and query $y(Az - sw)$.
    Compute $\hat{p}_s$ the empirical probability of $y(Az - sw) = -1$
    **if** $\hat{p}_s > p(b, s)/3$ **then**
        Count $\leftarrow$ Count+1
**if** Count $> 3T/4$ **then**
    **return** "Yes"
**else return** "No"

---

*Proof of Lemma F.3.* By Lemma D.3 and Lemma F.2, we know that over the randomness of $s$, with probability at least $5/6$, $\eta := \mathbf{Pr}_{z \sim N(0, I)}(h^*(Az - sw) \neq y(Az - sw)) \leq 6\epsilon \exp(s^2/2)/b$. We assume, for now, such an event happens. We first show that such a noise rate is much smaller than $p(b, s)$. Write $s = at + \xi$, where $\xi \in [0, b]$, then we have

$$
\frac{\epsilon \exp(s^2/2)/b}{\frac{1}{T_{bs}} \exp(-T_{bs}^2/2)} = \frac{T_{bs}\epsilon}{b} \exp(\frac{s^2 + T_{bs}^2}{2}) \leq t\epsilon \exp(\frac{s^2}{2} + \frac{(t - as)^2}{2b^2})
$$

$$
\leq t(t \exp(t^2/2))^{-1} \exp(\frac{s^2}{2} + \frac{(t - as)^2}{2b^2})/C
$$

$$
= C^{-1} \exp(-\frac{t^2}{2} + \frac{(at + \xi)^2}{2} + \frac{(b^2 t - a\xi)^2}{2b^2})
$$

$$
= C^{-1} \exp(\frac{1}{2}(\xi^2 + \frac{a^2 \xi^2}{b^2})) \leq C^{-1} e := (C')^{-1}, \tag{3}
$$

where, in the first inequality, we use the fact that $T_{bs} \leq bt$, in the second inequality, we use the fact that $t \exp(t^2/2) \leq 1/(C\epsilon)$ for a sufficiently large constant $C$, and in the last inequality, we use the fact that $a^2 + b^2 = 1, \xi^2 < b^2$. By Fact D.4, we know that $\exp(-T_{bs}^2/2)/T_{bs}$ is at most 3 times $p(b,s)$, and thus $\eta \leq p(b,s)/C'$ for a large enough constant $C'$.

By Fact 3.3, we know that the ground truth label $\ell(z) = h^*(Az - sw) = \text{sign}((a^*\sigma w + b^* u) \cdot z + t^* - a^* s)$. By Hoeffding's inequality, with high probability, we are able to estimate the probability of $y(Az - sw) = -1$ up to error $p(b,s)/20$ using $\tilde{O}(1/p^2(b,s))$ queries. In particular, since $T_{bs} \leq tb < 1/4$, by Fact D.4, we know that $p(b,s) > p^{1/4}$ and will cost us only $\tilde{O}(1/\sqrt{p})$ queries.

If the probability that $p_s > p(b,s)/4$ is at most $1/2$, then in each round $i$ of Algorithm 4, with probability at least $1/3$ it holds simultaneously that $p_s < p(b,s)/4$ and $\eta \leq p(b,s)/C'$. In this case, with high probability $\hat{p}_s < p(b,s)/3$ and Count does not increase. Thus, with probability at least $1 - \delta$, after $T = O(\log(1/\delta))$ rounds, Count $< 3T/4$ by Hoeffding's inequality.

Similarly, if the probability that $p_s > p(b,s)/2$ is at least $29/30$, then in each round $i$ of Algorithm 4, with probability at least $4/5$ it holds simultaneously that $p_s > p(b,s)/2$ and $\eta \leq p(b,s)/C'$. In this case, with high probability $\hat{p}_s > p(b,s)/3$ and Count increases. Thus, with probability at least $1 - \delta$, after $T = O(\log(1/\delta))$ rounds, Count $> 3T/4$ by Hoeffding's inequality.

Finally, we show that when $|b^* - b| \leq 1/\log(1/\epsilon)$ and when $|t - t^*| \leq 1/\log(1/\epsilon)$, Algorithm 4 with high probability outputs "Yes". To do this, we will show the true threshold of $\ell(z)$ is close to $T_{bs}$. We have

$$\frac{t^* - a^* s}{\sqrt{(b^*)^2 + (a^*\sigma)^2}} - T_{bs} \leq \frac{t^* - a^* s}{b^*} - \frac{t - as}{b} \leq \frac{O(\log(1/\epsilon)^{-1})}{b^*} + |t - as| \left| \frac{1}{b^*} - \frac{1}{b} \right|$$

$$\leq O(\log(1/\epsilon)^{-1/2}) + \left| \frac{b^2 t(b - b^*)}{b^* b} \right|$$

$$= O(\log(1/\epsilon)^{-1/2}) + O(t \log(1/\epsilon)^{-1}) = O(\log(1/\epsilon)^{-1/2}).$$

By Fact D.4, it holds with probability 1 that $p_s > p(b,s)/2$.

$\square$

Now assume that we have $\sin(\theta_i/2) \leq \sigma_i$, then $b_i \leq 2\sigma_i$. This implies that by testing $b = 2\sigma_i - \frac{j}{\log(1/\epsilon)}$ for $j = 0, 1, \ldots$, we only need $O(\log(1/\epsilon))$ rounds to find the correct $b$. With this fact, we have the following Algorithm 5.

*Proof of Theorem F.1.* Let $\theta_i = \theta(w_i, w^*)$ and write $w^* = a_i w_i + b_i u_i$, where $a_i, b_i > 0, a_i^2 + b_i^2 = 1$. By Algorithm 2, we know that with probability at least $1/3$, $\sin(\theta_0/2) \leq O(\epsilon/p\sqrt{\log(p/\epsilon)})$. We will assume $\sin(\theta_0/2) \leq \epsilon/p\sqrt{\log(p/\epsilon)}$ holds throughout the proof, since the constant before $\epsilon/p\sqrt{\log(p/\epsilon)}$ can always be assumed to be normalized as $1/C$ is large enough. In round $i$ of the algorithm, we write $w^* = a_i w_i + b_i u_i$ where $a_i, b_i > 0, a_i^2 + b_i^2 = 1$. Similar to the analysis of Algorithm 3, we will show that if $\sin(\theta_i/2) \leq \sigma_i$ then with probability $1/3$ it also holds that $\sin(\theta_{i+1}/2) \leq \sigma_{i+1}$. If this is true then since $1/t > 1/\sqrt{\log(1/\epsilon)}$ after $O(\log\log(1/\epsilon))$ rounds, we have $\sin(\theta_T/2) \leq 1/t$ with probability at least $1/\text{polylog}(1/\epsilon)$.

Recall the notation in the proof of Lemma F.3. Given $\hat{b}$, we define $p(\hat{b}, s)$ to be the bias of a halfspace with a threshold $T_{\hat{b},s} = (t - \hat{a}s)/\hat{b}$. By Fact 3.3, we define $\ell(z) = h^*(Az - sw_i) = \text{sign}((a_i\sigma w_i + b_i u) \cdot z + t^* - a_i s)$ the ground truth label of $y(Az - sw_i)$, $t_s$ to be its threshold and $p_s$ to be the bias of $\ell(z)$.

By Lemma F.3, we know that as long as $b_i > 1.5/t$, with high probability Algorithm 4 will output "Yes" for some $\hat{b}$ such that with probability at least $1/2$, $p_s > p(\hat{b}, s)/2 > p^{1/4}$. On the other hand, by Equation (3), we know that with probability at least $5/6$, $\eta := \mathbf{Pr}_{z \sim N(0,I)}(\ell(z) \neq y(Az - sw_i)) \leq p(b,s)/C'$ for a sufficiently large constant $C'$. Thus, with a probability at least $1/3$, $p_s > p(\hat{b}, s)/2$ and $\eta \leq p(\hat{b}, s)/C'$ hold simultaneously. For now, we assume this happens and we will analyze the smoothed label around some $z_0$ such that $y(Az_0 - sw_i) = -1$. By Fact 3.10, the smoothed label

---

**Algorithm 5** INITIALIZATION 2(Finding a good initialization under extreme threshold)

---

**Input:** error parameter $\epsilon \in (0,1)$, confidence parameter $\delta \in (0,1)$, threshold $t > 0$
**Output:** $w_0 \in S^{d-1}$
Run Algorithm 2 to get a $w_0 \in S^{d-1}$. Let $\sigma_0 = \epsilon/p\sqrt{\log(p/\epsilon)}$ be a parameter
**for** $i = 0, \ldots, T = O(\log\log(1/\epsilon))$ **do**
    Run Algorithm 4 with input $w_i$ and $\hat{b} = 2\sigma_i - \frac{j}{\log(1/\epsilon)}, j = 0, \ldots, \sigma_i \log(1/\epsilon)$
    Let $\hat{b}$ be the first parameter such that Algorithm 4 outputs "Yes"
    If Algorithm 4 outputs "No" for all $\hat{b}$ or the $\hat{b}$ we use less than $1/t$, then **return** $w_i$.
    Let $\hat{a} = \sqrt{1 - \hat{b}^2}$ and $T_{\hat{b},s} = (t - \hat{a}s)/\hat{b}$, where $s \sim [\hat{a}t, \hat{a}t + \hat{b}]$.
    Estimate the probability $\hat{p}_s$ of $p_s = y(Az - sw_i) = -1$ for $z \sim N(0, I)$ up to error $p^{1/4}/100$
using $\tilde{O}(\sqrt{1/p})$ queries. Let $\hat{t}_s$ be the threshold of a halfspace with bias $\hat{p}_s$
    $A \leftarrow I - (1 - \sigma^2)w_i w_i^t, \sigma = 1/t$
    Draw $z \sim N(0, I)$ and query $y(Az - sw_i)$ until some $z_0$ such that $y(Az_0 - sw_i) = -1$ is
drawn
    Draw $z_i \sim N(0, I)$, for $i \in [m], m = \tilde{O}(d)$ and query $f_i(z_i) := y(A(\sqrt{1 - \rho}z_0 + \rho z_i) - sw_i)$,
where $\rho = 1/\hat{t}_s$
    $g_i \leftarrow \frac{1}{m}\sum_{i=1}^m \text{proj}_{w_i^\perp} z_i f_i(z_i), w_{i+1} \leftarrow \text{proj}_{S^{d-1}}(w_i + \mu_i g_i)$
    $\sigma_{i+1} \leftarrow (1 - 1/C_2)\sigma_i \, \mu_{i+1} = (1 - 1/C_1)\sigma_{i+1}$
**return** $w_T$

---

around $z_0$ with respect to halfspace $\ell(z)$ can be seen as a halfspace

$$\ell_{z_0}(z_i) = \text{sign}(W_i \cdot z_i + T_{\rho,s}),$$

where $W_i := (a_i \sigma w_i + b_i u_i)/\sqrt{(a_i\sigma)^2 + b_i^2}$ and $T_{\rho,s} = \frac{t_s + \sqrt{1-\rho^2}W_i \cdot z_0}{\rho}$

By Fact 3.10 and Lemma 3.11, we know that with probability at least $1/2$, such a $z_0$ satisfies

1. $W_i \cdot z_0 \in (-t_s - 1/t_s, -t_s)$

2. the noise level of the smoothed label is at most $5\eta/p_s \leq 1/C''$ for some large enough constant $C''$

Since $p_s > p(\hat{b}, s)/2$, we can bound the threshold $T_{\rho,s}$ by

$$-2 \leq \frac{t_s - \sqrt{1-\rho^2}(t_s + \frac{1}{t_s})}{\rho} \leq T_{\rho,s} \leq \frac{t_s(1 - \sqrt{1-\rho^2})}{\rho} \leq t_s/\hat{t}_s \leq 2, \quad (4)$$

because $\hat{t}_s$ is at least close to $t_s$ up to a small constant factor, otherwise $\hat{p}_s$ would be far from $p_s$. Combine Equation (4) and Fact 3.4, we know that $\mathbf{E}_{z' \sim N(0,I)} \text{proj}_{w_i^\perp} z' \ell_{z_0}(z') = \phi \frac{u_i b_i}{\sqrt{(a_i\sigma)^2 + b_i^2}}$, for some $\phi \in (e^{-2}, 1)$. Since the noise level of the smoothed label around $z_0$ is as small as $1/C''$ for some large enough constant $C''$, by Hoeffding's inequality, we know that $\left\| g_i - \mathbf{E}_{z' \sim N(0,I)} \text{proj}_{w_i^\perp} z' \ell_{z_0}(z') \right\|$ can be smaller than some tiny constant with high probability.

As it always holds that $\sigma_i \geq 1/t$ for each $i$, we will consider two cases. In the first case, $\sin(\theta_i/2) \leq 3\sigma_i/4$ and $\|g_i\|$ is bounded by some universal constant.

In the second case, we have $3\sigma_i/4 \leq \sin(\theta_i/2) \leq \sigma_i$. In this case we know that $\mathbf{E}_{z' \sim N(0,I)} \text{proj}_{w_i^\perp} z' \ell_{z_0}(z') = \phi \frac{u_i b_i}{\sqrt{(a_i\sigma)^2 + b_i^2}} = \psi u_i$ for some $\psi \geq e^{-4}$, which implies that $\mathbf{E}_{z' \sim N(0,I)} \text{proj}_{w_i^\perp} z' \ell_{z_0}(z') \cdot u_i \geq \psi \frac{b_i}{\sqrt{(a_i\sigma)^2 = b_i^2}} \geq e^{-5}$. Using Lemma 3.2, we know that $\sin(\theta_{i+1}/2) \leq (1 - 1/C_1)\sigma_i = \sigma_{i+1}$.

Finally, we prove the query complexity of Algorithm 5. By Theorem 3.8, it takes us $\tilde{O}(1/p + d\log(1/\epsilon))$ queries to get some $w_0$ by running Algorithm 2. After obtaining $w_0$, in each round of

Algorithm 4, we will run Algorithm 4 $O(\log(1/\epsilon))$ times to find a desired $\hat{b}$ and each round takes us $\tilde{O}(1/p^2(\hat{b})) \leq 1/p^{2c} \leq 1/\sqrt{p}$ queries, because $p(\hat{b})$ is the bias of a halfspace with threshold $T_{\hat{b}} = \hat{b}t$, which is smaller than $t$ by a tiny constant factor. Furthermore, after obtaining $\hat{b}$ it takes us $\tilde{O}(1/p(\hat{b}) + d\log(1/\epsilon))$ queries to perform the gradient descent update. So, in total Algorithm 5 has query complexity at most $\tilde{O}(1/p + d\log(1/\epsilon))$.

$\square$

# G   Proof of Theorem 1.2

*Proof of Theorem 1.2.* We first show the correctness of Algorithm 1. When we run Algorithm 1, we will start with some interval $[t_a, t_b]$ such that any halfspace with a threshold $t \in [t_a, t_b]$ must have bias $\Theta(p)$. Next, Algorithm 1 partition $[t_a, t_b]$ into grid such that $|t_{j+1} - t_j| \leq 1/\log(1/\epsilon)$. This implies that there must be some $t_j \in [t_a, t_b]$ such that $t_j - 1/\log(\log(1/\epsilon)) \leq t^* \leq t_j$. By Theorem 3.8 and Algorithm 5, as long as $p > C\epsilon$, with probability at least $1/\mathrm{polylog}(1/\epsilon)$, we can find some $w_0$ such that $\sin(\theta_0/2) \leq \min\{1/t_j, 1/2\}$. In particular, by running Algorithm 2 or Algorithm 5 $\mathrm{polylog}(1/\epsilon)$ times, at least one of these $w_0$ satisfies the condition. Furthermore, with such a $w_0$, we know from Theorem 3.1 that we can with high probability get some $\hat{h}$ such that $\mathrm{err}(\hat{h}) \leq O(\mathrm{opt} + \epsilon)$. Thus within the list $\mathcal{C}$ of the candidate hypotheses maintained by Algorithm 1 at least one of them has error $O(\mathrm{opt} + \epsilon)$. By Lemma C.1, we can with high probability find a hypothesis among $\mathcal{C}$, whose error is at most 10 times the error of the best hypothesis in $\mathcal{C}$ and thus has error $O(\mathrm{opt} + \epsilon)$.

Next, we prove the query complexity of Algorithm 1. According to Appendix C.2, we know that finding an interval $[t_a, t_b]$ costs us $\tilde{O}(\min\{1/p, 1/\epsilon\})$ queries. If we find $p < C\epsilon$ then we are done. Otherwise, we will run the initialization algorithm and the refinement algorithm. By Theorem 3.8 and Algorithm 5, each time we run an initialization algorithm, it takes us $\tilde{O}(1/p + d\,\mathrm{polylog}(1/\epsilon))$ queries. By Algorithm 3, each time we run Algorithm 3, it takes us $\tilde{O}(d\log(1/\epsilon))$ queries. Since we will run these algorithms at most $\mathrm{polylog}(1/\epsilon)$ times. We will in total make $\tilde{O}(1/p + d\,\mathrm{polylog}(1/\epsilon))$ queries. Finally, by Lemma C.1, finding a good hypothesis from the list of candidate hypotheses will only take us $\mathrm{polylog}(1/\epsilon)$ queries. Thus, we conclude the query complexity of Algorithm 1 is $\tilde{O}(\min\{1/p, 1/\epsilon\} + d\,\mathrm{polylog}(1/\epsilon))$.

$\square$

# H   Implementing the Learning Algorithm via A Small-Class Oracle

In this section, we discuss how to implement Algorithm 1 to get an even smaller query complexity $\tilde{O}_\delta(d \cdot \mathrm{polylog}(1/\epsilon))$, assuming there is an oracle that can return a random small-class example. Before presenting the definition of the small-class oracle, we remind the reader that the notation $\tilde{O}$ hides the dependence on $\mathrm{polylog}(1/\epsilon)$, and the notation $O_\delta$ hides the dependence on $\mathrm{polylog}(1/\delta)$. A small class oracle is defined as follows.

**Definition H.1** (Small-Class Oracle). *Let $D$ be a distribution over $\mathbb{R}^d \times \{\pm 1\}$ and $h^* = \mathrm{sign}(w^* \cdot x + t^*)$, $w^* \in S^{d-1}, t^* > 0$ be an optimal halfspace such that $\mathrm{err}(h^*) = \mathrm{opt} = \min_{h \in H} \mathrm{err}(h)$. A small-class oracle $EX^{(-)}(D)$ draws $(x, y) \sim D\,|_{y=1}$ and returns $x$.*

In other words, a small-class oracle simulates the following rejection sampling procedure, where a learner keeps drawing $x \sim N(0, I)$, querying its label and stops when it sees some $x_0$ with $y(x_0) = -1$. Such a procedure requires $\Omega(1/p)$ queries to implement, which is costly when $p$ is small.

By Theorem 3.1, even without the small-class oracle, the query complexity of Algorithm 3 is always $\tilde{O}_\delta(d \cdot \mathrm{polylog}(1/\epsilon))$. Thus, a small-class oracle would only help reduce the query complexity of Algorithm 2 and Algorithm 5. In the rest of the section, we show that by calling the small-class oracle $\tilde{O}_\delta(1) = O_\delta(\mathrm{polylog}(1/\epsilon))$ times, we can reduce the query complexity of Algorithm 2 and Algorithm 5 to $\tilde{O}_\delta(d \cdot \mathrm{polylog}(1/\epsilon))$.

We first consider Algorithm 2. By Theorem 3.8, the query complexity of Algorithm 2 is $\tilde{O}(1/p + d\log(1/\epsilon))$, where Line 3 in Algorithm 2 takes $1/p$ queries to find a random small-class example and Line 4-Line 5 in Algorithm 2 takes $\tilde{O}(d\log(1/\epsilon))$ queries. As a small-class oracle simulates the same rejection sampling procedure as Line 3 in Algorithm 2, we can implement Line 3 in Algorithm 2 with a single small-class oracle. Thus, by a single call of the small-class oracle, we are able to implement Algorithm 2 with $\tilde{O}_\delta(d \cdot \mathrm{polylog}(1/\epsilon))$ query complexity.

Next, we consider Algorithm 5. Each implementation of Algorithm 5 runs in $O(\log\log(1/\epsilon))$ iterations. In each iteration, we call Algorithm 4 $\mathrm{polylog}(1/\epsilon)$ times in Line 5, use queries to estimate $\hat{p}_s$ in Line 9, find a single-small class example in Line 11 and improve the current hypothesis with $\tilde{O}(d)$ queries in Line 12. Furthermore, only operations in Line 5, Line 9, and Line 11 have query complexity much larger than $\mathrm{polylog}(1/\epsilon)$. Thus, we only need to show with a small-class oracle, we can significantly reduce the query complexity of these steps.

We start with Algorithm 4. In Line 9 in Algorithm 4, we use query to estimate the probability of $y(Az - sw) = -1$ with an error up to error $p(b, s)$. By Lemma D.3, we know that if we pass a random sample $x \sim N(0, I)$ to the $(w, s, \sigma)$-rejection procedure, then the resulting distribution is $N(-sw, A)$. Thus, the probability of $y(Az - sw) = -1$ is exactly equal to the fraction of negative examples among examples that pass the $(w, s, \sigma)$-rejection procedure. Specifically, for the $(w, s, \sigma)$-rejection procedure, we denote by $q$ the probability that a random example passes the rejection procedure and denote by $q_-$ the probability that a random negative example passes the rejection procedure and $p$ the fraction of the negative example. Then we have $\mathbf{Pr}_{z \sim N(0,I)}(y(Az - sw) = -1) = pq_-/q$. This implies that if we know $p$ and $q_-$, then estimating $\mathbf{Pr}_{z \sim N(0,I)}(y(Az - sw) = -1)$ is equivalent to estimating $q_-$, which can be done by calling the small-class oracle several times and estimate the probability that these examples pass the $(w, s, \sigma)$-rejection procedure. By Lemma D.3, we know that $\sigma\exp(-s^2/(2(1 - \sigma^2)))$ can be computed precisely using the parameter $s, \sigma$. However, we do not know $p$ precisely, as this requires us to know $t^*$ up to a high accuracy. To overcome this difficulty, we use $\hat{p}$, the bias of a halfspace with threshold $t$, because we only need to ensure the correctness of the algorithm when our guess $t$ is close to $t^*$. In fact, when $|t - t^*| \le 1/\log(1/\epsilon)$, $\hat{p} \in [(1 - 1/C)p, (1 + 1/C)p]$ for some large enough constant $C$, which is enough for ensuring the correctness of Algorithm 4. So, to estimate $\mathbf{Pr}_{z \sim N(0,I)}(y(Az - sw) = -1)$ up to error $p(b, s)$, we only need to estimate $q_-$ up to error $qp(b,s)/\hat{p}$. We have

$$\frac{p(b,s)q_-}{\hat{p}} \ge \Omega\left(\frac{\sigma\exp(-s^2/2)\frac{1}{T_{bs}}\exp(-T_{bs}^2/2)}{\frac{1}{t}\exp(-t^2/2)}\right) \ge \Omega\left(\frac{1}{T_{bs}}\exp\left((t^2 - s^2 - T_{bs}^2)/2)\right)\right)$$

$$\ge \Omega\left(\frac{1}{T_{bs}}\exp\left((t^2 - s^2 - T_{bs}^2)/2)\right)\right) = \Omega\left(\frac{1}{T_{bs}}\exp\left((t^2 - (at + \xi)^2 - \left(\frac{t - a(at + \xi)}{b}\right)^2/2)\right)\right)$$

$$= \Omega(1/T_{bs}) \ge \Omega(1/\log(1/\epsilon)), \tag{5}$$

where we use Fact D.4 and $s = at + \xi, \xi \in [0, b]$. This implies that we only need to call a small class oracle $\tilde{O}_\delta(1) = O_\delta(\mathrm{polylog}(1/\epsilon))$ times to estimate $q_-$ and thus can compute $\mathbf{Pr}_{z \sim N(0,I)}(y(Az - sw) = -1)$ up to error $p(b, s)$. In particular, in this implementation, we only need to call the small-class oracle and do not need to make membership queries.

Similarly, to implement Line 9 in Algorithm 5, we also only need to call the small-class oracle $\tilde{O}_\delta(1)$ times and do not need to make membership queries.

Finally, we show that by calling the small-class oracle $\tilde{O}_\delta(1)$ times, we are able to implement Line 11 in Algorithm 5. By Lemma D.3, Line 11 in Algorithm 5 draws a random negative example that passes the $(w_i, s, \sigma)$-rejection procedure. By Lemma F.3, we know that $p_s = pq_-/q \ge \Omega(p(b, s))$. This implies that $q_- \ge \Omega(p(b, s)q/p) \ge \Omega(1/\log(1/\epsilon))$, by Equation (5). Thus, with high probability, we only need to pass $O_\delta(\log(1/\epsilon))$ examples from the small class oracle to the $(w_i, s, \sigma)$-rejection procedure to get one negative example that passes this rejection procedure. Thus, in each iteration of Algorithm 5, we will call $\tilde{O}_\delta(1)$ times the small-class oracle and make $\tilde{O}_\delta(d)$ membership queries.

In summary, we count the number of queries in Algorithm 1 using the new implementation with a small class oracle. Notice that with a small-class oracle, we do not need to worry about using some guess $t$ much larger than $t^*$ because the query complexity in the initialization step now has no dependence on the bias of the target halfspace. So we do not need to implement line 4 in Algorithm 1

but only need to guess $t' = i/\log(1/\epsilon)$ for $i = 0, \ldots, \lceil \text{polylog}(1/\epsilon) \rceil$. This means in Algorithm 1, we will call Algorithm 2 and Algorithm 5 in total at most $\text{polylog}(1/\epsilon)$ times, so we will make $\tilde{O}_\delta(1)$ small-class oracles and make $\tilde{O}_\delta(d \cdot \text{polylog}(1/\epsilon))$ membership queries.

