# OpenReview forum: "Active Learning of General Halfspaces: Label Queries vs Membership Queries"
_NeurIPS.cc/2024/Conference — NeurIPS 2024 poster_

### Official Review · Reviewer_vQ1c · 2024-06-13

**Soundness:** 4
**Presentation:** 3
**Contribution:** 3
**Rating:** 7
**Confidence:** 4

**Summary:**

This paper studies the problem of actively learning non-homogeneous half-spaces under Gaussian distribution. It shows that in the pool based model, any active learner cannot do better than the passive learner in terms of label complexity, unless exponentially many unlabelled examples are drawn. On the other hand, it shows that in the membership query model, there exists active learner that can outperform the passive learner, thus demonstrating a separation between these two models.

**Strengths:**

1. This paper settled down the open problem whether it is possible to learn non-homogeneous half-spaces using $O(d\log(1/\epsilon)+\min(1/p,\epsilon))$ labelled data and $O(d/\epsilon)$ unlabelled data under the Gaussian distribution in the pool-based active learning setting. It then shows that this label complexity is achievable under the membership query model, thus there's a separation of the two models. This result and technique used is complete and novel.

2. This paper is well-organized. It gives enough motivation and background information to the problem and discussed related work.

3. The writing is very clearly for most of the part. Notations are well-defined and the math is rigorous.

**Weaknesses:**

1. (I would adjust my rating based on the response of this point). The only real issue I have here is that I'm not fully convinced by your conclusion that "in the pool-based model, active learning has no improvements over passive unless the pool size is exponential". The passive bound has an explicit dependence on $\epsilon$ while your Theorem 1.1 doesn't. It only has a dependence on $1/p$. If $p$ is constant, your conclusion is not correct. So your conclusion only holds for some regime of $p$. This also makes me wonder how tight this lower bound is, is it possible to have a lower bound with $\epsilon$ appearing explicitly here?

2. The lower bound holds for the realizable case while the upper bound holds for the upper bound is for the agnostic setting. This leaves a gap in the realizable upper bound and potentially easier algorithms.

3. The paper is getting a bit too technical at the end. All the writings are extremely well, especially the lower bound part, but it gets more technical in section 3 and it's hard to follow for me.

     i) I think it's better to give some examples in section 3, explaining the intuition more and moving some of the lemmas to appendix.

     ii) It feels like your MQ algorithm isn't completely new but used the framework/idea/techniques (like the localization) from previous works. It's better to start with a simpler algorithm and explain what's new in your paper.

     iii) It might be better to have a conclusion section to discuss some future directions.

4. There're some very tiny typos.

   i) In the abstract, your lower bound is $\Omega(d/\log(m)\epsilon)$ while in your theorem it is $\Omega(d/\log(m)p)$.

   ii) In algorithm 1, what's the subroutine TEST C? In the appendix you have a subroutine named ANGLE TEST.

   iii) Line 337, you typed "the length of" twice.

**Questions:**

1. As I mentioned, is there a better MQ bound or simpler algorithm in the realizable case? In the realizable case is it possible to get $\text{OPT}+\epsilon$ exactly?

2. Is there some distribution that is natural and actively can have an advantage over passive learning? I still believe active learning would have some advantages in many instances over passive for learning non-homogenous half-space in the pool-based setting.

3. On the other hand, is it easy to extend your result to more general distributions? Like log-concave distributions?

**Limitations:**

It's better to have a conclusion section to talk about future directions.

---

> ### Author Rebuttal · Authors · 2024-08-07
>
> We want to thank the reviewer for appreciating our work and providing useful suggestions. We next respond to the comments from the reviewer as follows.
>
> ## The benefit of pool-based active learning:
> We start by commenting on the statement of Theorem 1.1. In the statement, we show that to learn a $p$-biased halfspace up to error $O(p)$, one need to make at least $d/p$ queries. Since $p \ge \epsilon$, this lower bound also holds if we want to learn a $p$-biased halfspace up to error $\epsilon$. On the other hand, $d \log(1/\epsilon)$ is the minimum amount of information to describe a halfspace up to error $\epsilon$. Thus, for $\epsilon\le p$, one can naively get a lower bound of $d/p+d \log(1/\epsilon)$. If $p$ is large, then of course pool-based active learning can beat passive learning. This has already been shown in the literature that studies learning homogeneous halfspaces. If we place no restriction on $p$, then in the worst case, we need to pay $d/\epsilon$ many label queries to learn an $\epsilon$-biased halfspace. Thus, the $d/\epsilon$ lower bound stated in the abstract is consistent with the statement of Theorem 1.1
>
> We next point out that the power of membership queries is used to obtain the warm start. Provided a warm start, the refinement step can be simulated using pool-based active learning efficiently (this can be verified based on Lemma D.3). So, if $p$ is not very small, we can take the average of roughly $\tilde{O}(d)$ small class examples and use the mean as a warm start; this takes $\tilde{O}(d/p)$ label queries. In this case, we obtain an efficient pool-based active learning algorithm with $d/p+d\log(1/\epsilon)$ label complexity, which gives some benefit for learning balanced halfspaces. However, such a benefit vanishes when $p$ approaches  $\epsilon$.
>
> ## Potential simple algorithm in the realizable setting using MQ:
> We first note that in the realizable setting our algorithm also achieves error $opt+\epsilon$ (since $opt=0$). Furthermore, we want to mention that membership query learning can be too strong for distribution-specific learning in a realizable setting as the problem becomes a geometric problem and can sometimes be trivial. This is one reason why membership query is widely studied in the noisy setting instead. Recall that membership query is defined over the support of the marginal distribution. Assuming the underlying distribution is uniform distribution over the unit sphere instead, one can first pay $1/p$ queries on the sphere to get an example from the small class. Based on this example, one can do $d$ times binary searches to find $d$ examples $\epsilon$-close to the decision boundary. This gives a simple learning algorithm in the realizable setting with an optimal query complexity $\min\(1/p,1/\epsilon)+d\log(1/\epsilon)$. If the distribution is Gaussian, the support is the whole space instead. As we mentioned in line 88-91, one can query examples that are very far from the origin to easily get a small class example and thus avoid the overhead term $1/p$. In such a simple learning algorithm, many queries are made over regions with a very small probability mass, which makes the algorithm very fragile. This is one of the reasons why we consider agnostic learning in this paper. Since in high dimension, Gaussian well approximates the uniform distribution over a sphere with radius $\sqrt{d}$, if we add an exponentially small level of label noise, then queries outside such a sphere provide no useful information to the learner.
>
> ## Overview of novel ideas and techniques:
> We will add such an overview to intuitively explain the new techniques developed in the paper. It should be emphasized that even though we use some form of localization (which is a previously developed technique) to refine a warm start, to match the optimal query complexity, several new ideas are required. Intuitively, even to naively estimate the bias of the target function, one roughly needs $1/\epsilon^2$ many queries. Surprisingly, we managed to learn the direction and the bias of the target halfspace simultaneously with a low query complexity. Moreover, even in the localization step, new ideas are needed to handle the issue that the true threshold is unknown (as previous work mainly handles the homogeneous case). Besides the ideas about localizations, we also introduced novel ideas to design robust algorithms to obtain a good initialization with a low query complexity. Before this work, initializations are usually obtained by estimating Chow-parameters and need significantly more queries. Furthermore, the techniques we developed for proving our lower bound give the first improvevement over the $1/p$ bound by [Das04]; we believe the ideas of the proof can be applied to other problems.
>
>
> ## Distributional assumption:
> We start by pointing out that this is the first work that studies the label complexity of learning general halfspaces; focusing on the basic setting that the data follows the Gaussian distribution is an important first step in this direction. For more general structured distributions, we believe that the techniques (such as the localization for general halfspaces) used in our algorithm could be extended to more general marginal distributions. However, to achieve optimal label complexity, more careful analysis is needed. It is worth mentioning that even in the passive learning setting, efficient algorithms for agnostically learning general halfspaces are only known under the Gaussian distribution. Thus, we expect that there will be follow-up works that study the optimal label complexity of learning general halfspaces under more general distributional assumptions such as isotropic log-concave distributions. Finally, we would like to refer the reviewer to our response to Reviewer dMkP which answers this question in more detail.

---

> > ### Comment · Reviewer_vQ1c · 2024-08-09
> >
> > Thank you for your response, I adjusted my rating.

---

### Official Review · Reviewer_88gR · 2024-07-01

**Soundness:** 3
**Presentation:** 2
**Contribution:** 3
**Rating:** 7
**Confidence:** 2

**Summary:**

This paper first provides a lower bound on active learning using label queries. This lowerbound is nearly tight compared with upperbound for label queries. To get around the lowerbound, the authors study active learning using membership query, which means the algorithm can directly access the random function that gives the label y to x. The membership query is clearly stronger than label queries and hence, the authors manage to prove an upperbound. The authors also mentioned lowerbounds in membership query from other work.

**Strengths:**

The settings of active learning using label query and membership query are very natural and well-studied. The authors manage to provide a nearly-tight lowerbound and an interesting upperbound for general halfspace. The strategy they use in both proofs like considering negative examples in samples for lowerbound and finding a subclass using good initialization and fine-tuning seems to be intuitive and interesting.

**Weaknesses:**

I think the results are very interesting but the paper writing can be improved.

**Questions:**

How do you know whether to use initialization for an extremely large threshold(section F) or not (section E)?
I think it would make the paper easier to read if the authors could describe what 3.1 is doing at high level.
I think there is a small notation error in A.2(sample (x,y)\sim D instead of N(0, I)).

---

> ### Author Rebuttal · Authors · 2024-08-07
>
> We want to thank the reviewer for appreciating our work and providing constructive feedback. We will improve the writing in the revised version of this work. Below, we answer the question from the reviewer.
>
> ## Using the correct initialization algorithm:
>
> This is a good question: how to estimate the true threshold plays a very important role of controlling the query complexity. In fact, we do not need to know such a threshold at the beginning; we guess such a threshold up to an additive $1/\log(1/\epsilon)$ error and use the guess to determine which initialization algorithm to use. The guessed threshold will be refined in the learning process.

---

> > ### Comment · Reviewer_88gR · 2024-08-10
> >
> > Thank you for your explanation. I will keep my score as 7.

---

### Official Review · Reviewer_goh3 · 2024-07-01

**Soundness:** 3
**Presentation:** 2
**Contribution:** 2
**Rating:** 5
**Confidence:** 3

**Summary:**

The paper studies the question of active learning halfspaces over $\mathbb{R}^d$, under the Gaussian distribution in two different models: Label queries and membership queries. In the label queries model, the authors prove a lower bound implying that the learner must have a pool of size $exp(d)$ in order to do better (in terms of samples/queries) than a passive learner. In the membership queries model, the authors prove that the above lower bound can be circumvented by using the strength of membership queries, and provide an upper bound which is better than the optimal sample complexity bound of passive learners. The upper bound is given by an efficient semi agnostic learner (a learner with risk guarantee of the form $O(opt + \epsilon)$).

**Strengths:**

1. The studied problem is of a fundamental nature.
2.  The lower bound and its circumvention via membership queries are interesting.
3. The upper bound algorithm is efficient.

**Weaknesses:**

1. I find the introduction not clear enough. For example, I do not understand how the upper bound in line 56 settles with the lower bound in line 54. The upper bound has no dependence on $1/\epsilon$. Also, how does Theorem 1.1 relate to those bounds? The bounds relate to the uniform distribution while the theorem relates to the Gaussian distribution.
2. Line 93 says: "The overhead term cannot be avoided in the agnostic setting...". It is not discussed whether it can be avoided in the realizable setting.
3. While the results are potentially interesting, I think that the presentation of the paper needs to be improved. High level ideas and technical details are mixed, which makes the paper hard to follow.
4. No discussion on possible future research.
5. The results are applicable to a relatively limited setting of halfspaces under the Gaussian distribution.
6.  While the upper bound algorithm is efficient, it only gives a "semi-agnostic" guarantee in the form of $O(opt + \epsilon)$. However, the constant hidden in the $O$ notation is not specified (at least not in the theorem), and it can sometimes be important. In some scenarios it might be perfectly reasonable that $opt = 1/1000$. So, if the hidden constant is at least $500$, the guarantee is no better than random guessing in those cases. It could be interesting to design an inefficient learner with better risk guarantees.

**Questions:**

Questions:

1. In line 157, I don't understand what does it mean that an argument is "hard to formalize". Do you mean that it is counter-intuitive?
3. In line 205, do you mean that an upper bound $\epsilon$ on the noise level is *given* to the learner beforehand?


Suggestions:
1.  I would recommend writing a "technical overview" section that separates the high-level novel ideas from known techniques and technical details.
2. Add a discussion on future research. For example, in Theorem 1.2: can we do better in the realizable setting? Can we do better by using an inefficient learner?


Typos:
1. Line 337: "the length of the length of"
2. Line 389: "unfortinately:

**Limitations:**

Yes.

---

> ### Author Rebuttal · Authors · 2024-08-07
>
> We want to thank the reviewer for the feedback. We will revise the writing in the updated version of this manuscript. We will also include a section that discusses future directions. Below, we respond to the weaknesses and questions pointed out by the reviewer.
>
> ## Confusion in the introduction:
> >I do not understand how the upper bound in line 56 settles with the lower bound in line 54. The upper bound has no dependence on $1/\epsilon$
>
> The upper bound of $(1/p)d^{3/2}\log(1/\epsilon)$, achieved by an exponential time algorithm, is significantly worse than the naive lower bound of $\min(1/p,1/\epsilon)+d \log(1/\epsilon)$. Note that $1/p$ is at least $\min(1/p,1/\epsilon)$ and $d^{3/2}\log(1/\epsilon)$ is larger than $d \log(1/\epsilon)$.
> >How does Theorem 1.1 relate to those bounds? The bounds relate to the uniform distribution while the theorem relates to the Gaussian distribution.
>
> It is well-known that, in high-dimensions, the Gaussian distribution is well-approximated by the uniform distribution over the sphere with radius roughly $\sqrt{d}$. As a result, the learning problems under these two marginal distributions are almost equivalent. One could easily modify the calculation used in the proof of Theorem 1.1, to obtain the same lower bound for the uniform distribution over the unit sphere. Since the Gaussian distribution is arguably a more well-studied distribution in the literature (and the calculation is cleaner), we chose to study it instead of the uniform distribution over the unit sphere in this paper.
>
> >Line 93 says: "The overhead term cannot be avoided in the agnostic setting...". It is not discussed whether it can be avoided in the realizable setting.
>
> We included a brief discussion about why this term can be avoided in the realizable setting; see lines 88-91. Roughly speaking, the membership query model is fairly strong in the distribution-specific and realizable setting. This can render the learning task somewhat straightforward. Specifically, since the support of the Gaussian distribution is the whole Euclidean space, the learner can choose to query many points extremely far from the origin to quickly get small class examples; and then determine the decision boundary of the target halfspace using binary search. Such a naive method can avoid the overhead term and achieve a low label complexity. In contrast, if the support of the distribution is the unit sphere, then the $(\min\{1/p,1/\epsilon\}+d \log(1/\epsilon))$ lower bound also holds in the membership query setting; as the learner is not allowed to query outside the sphere. This is an additional motivation to study the agnostic setting, since an exponentially small level of noise can make points far from the origin provide no information. We refer the reviewer to our response to reviewer vQ1c for more details.
>
> ## Distributional assumption:
>
> We start by pointing out that this is the first work that studies the label complexity of learning general halfspaces; focusing on the basic setting that the data follows the Gaussian distribution is an important first step in this direction. For more general structured distributions, we believe that the techniques used in our algorithm could be extended to more general marginal distributions. However, to achieve optimal label complexity, more careful analysis is needed. It is worth mentioning that even in the passive learning setting, efficient algorithms for agnostically learning general halfspaces are only known under the Gaussian distribution. Thus, we expect that there will be follow-up works that study the optimal label complexity of learning general halfspaces under more general distributional assumptions. Finally, we would like to refer the reviewer to our response to Reviewer dMkP which answers this question in more detail.
>
> ## $O(\mathrm{opt}+\epsilon)$ guarantee:
> We want to point out that learning up to error $O(opt+\epsilon)$ is a standard benchmark --- many works can only achieve $O(\sqrt{opt}+\epsilon)$ or $O(opt\sqrt{\log(1/opt)}+\epsilon)$) in robust learning theory. There is a long line of works in this direction; see [ABL17,YZ17,DKS17] for some representative works. In particular, if $opt=0$, i.e., in the realizable setting, our algorithm learns the target hypothesis within error $\epsilon$ (for any desired $\epsilon>0$). As we mentioned in the introduction, previous work in the realizable setting uses an exponential time algorithm to learn up to error $\epsilon$ with a significantly sub-optimal query complexity. On the other hand, learning up to error $opt+\epsilon$ is known to be computationally intractable, even under the Gaussian distribution, as we mentioned in the introduction. This is why such a guarantee is usually not considered in this direction.
>
> >[ABL17] Awasthi, Pranjal, Maria Florina Balcan, and Philip M. Long. "The power of localization for efficiently learning linear separators with noise." Journal of the ACM (JACM) 63.6 (2017): 1-27.
>
> >[YZ17]Yan, Songbai, and Chicheng Zhang. "Revisiting perceptron: Efficient and label-optimal learning of halfspaces." Advances in Neural Information Processing Systems 30 (2017).
>
>
>
> ## Questions from the reviewer:
>
> >In line 157, I don't understand what does it mean that an argument is "hard to formalize". Do you mean that it is counter-intuitive?
>
> In line 147, we have provided intuition why such a lower bound makes sense. But it is hard to use formal mathematics to turn such an intuition into a proof. This is because we have to make analysis conditioned on very complicated probability events. On the other hand, the technique we used in this paper bypasses this difficulty.
> >In line 205, do you mean that an upper bound ϵ on the noise level is given to the learner beforehand?
>
> As we discussed in Appendix C.1, we can without loss of generality assume that $opt<\epsilon$; otherwise, we can use a standard doubling trick to guess such a noise level and repeat the algorithm $\log(1/\epsilon)$ times.

---

> > ### Comment · Reviewer_goh3 · 2024-08-08
> >
> > Thank you very much for addressing my comments and questions.
> > My main concern regarding this paper, as a very non-expert in active learning, was (and still is) it's inaccessible writing style.
> > I believe that the paper can highly benefit from adding a "technical overview" section, and from making another pass on it, while trying to read it from the eyes of a non-expert, and revise it accordingly. This is especially important if you "expect that there will be follow-up works".
> > However, since the authors commit to "revise the writing", I will update my final score.

---

### Official Review · Reviewer_amzj · 2024-07-11

**Soundness:** 3
**Presentation:** 3
**Contribution:** 3
**Rating:** 7
**Confidence:** 3

**Summary:**

This paper considers active learning of general (non-homogeneous) halfspaces under Gaussian distribution. Define p=P(Y=-1). On the one hand, it proves that, roughly speaking, with standard label queries, one cannot learn a classifier with O(opt+epsilon) error that requires polynomially many unlabeled samples and $O(d\log\frac{1}{\epsilon}+\text{min}(1/p, 1/\epsilon))$ labels. On the other hand, with membership queries, it gives a computationally efficient algorithm that achieves $O(d\text{poly}\log(\frac{1}{\epsilon})+\text{min}(1/p, 1/\epsilon))$.

**Strengths:**

- This paper considers an interesting open problem learning theory: a lot of research has considered learning homogeneous halfspaces, but what is the limit for non-homogeneous halfspaces? It gives an interesting result that it can't be very efficiently learned under the standard label query paradigm, even assuming gaussian distribution and noise-free, while membership query could help.

- The proposed method is sound and technically non-trivial, though I have not checked proofs in Appendix.

- The paper is written clearly in general, and explains high-level intuitions well.

**Weaknesses:**

- For the upper bound / algorithmic result, it looks to me it is highly depended on the Gaussian (or "nearly" symmetric) assumption. Does that mean that membership queries may not even be enough if the data distribution is more general?

- Also for the upper bound / algorithmic result, it aims at achieving O(opt+epsilon) instead of opt+epsilon error. I understand that in general the latter is proven to be hard, but it would be nice if there is a discussion about if the latter can be achieved with some assumptions about noise.

**Questions:**

N/A

---

> ### Author Rebuttal · Authors · 2024-08-07
>
> We want to thank the reviewer for appreciating our work and providing us with constructive feedback. We respond to the points made by the reviewer below.
>
> ## Strong marginal distributional assumption:
> We start by pointing out that this is the first work that studies the label complexity of learning general halfspaces; focusing on the basic setting that the data follows the Gaussian distribution is an important first step in this direction. For more general structured distributions, we believe that the techniques (such as the localization for general halfspaces) used in our algorithm could be extended to more general marginal distributions. However, to achieve optimal label complexity, more careful analysis is needed. It is worth mentioning that even in the passive learning setting, efficient algorithms for agnostically learning general halfspaces are only known under the Gaussian distribution. Thus, we expect that there will be follow-up works that study the optimal label complexity of learning general halfspaces under more general distributional assumptions such as isotropic log-concave distributions. Finally, we would like to refer the reviewer to our response to Reviewer dMkP which answers this question in more detail.
>
> ## Different noise models:
> The agnostic (or adversarial label noise) model used here is one of the strongest noise models in the literature.  A number of  pioneering prior works focused on this noise model in the context of passive and active learning; see, e.g., [ABL17,DKS17]. We believe that in weaker noise models, such as Random Classification noise, a similar algorithm could achieve error $opt+\epsilon$. We view this as a moderately interesting direction, as in such models one could make repeated queries to a single point and obtain the correct label (with high probability). On the other hand, membership query is defined over the support of the marginal distribution. As a result, many membership query learning algorithms take this advantage to learn by making many queries over regions with very small probability mass. For example, in the realizable setting, if the marginal distribution is Gaussian, then one can easily find small class examples by querying points that are extremely far from the origin, which makes the learning algorithm very fragile. This is also an important motivation for studying adversarial label noise. We can use adversarial label noise to model such a regime and restrict the power of query (an exponentially small level of noise suffices for this purpose). From this point of view, studying other types of label noise is beyond the scope of this paper. We will add these questions as future directions in a revised version of the manuscript.

---

> > ### Comment · Reviewer_amzj · 2024-08-12
> >
> > Thanks for the response. I will keep my score and support its acceptance.

---

### Official Review · Reviewer_dMkP · 2024-07-11

**Soundness:** 3
**Presentation:** 2
**Contribution:** 2
**Rating:** 6
**Confidence:** 3

**Summary:**

This paper studies the classical problem of distribution-dependent---here standard normal disitrbution---active learning of non-homogeneous halfspaces using label and membership queries in the realizable and agnostic setting. There are two main contributions.

1. A strong lower bound in the label query setting essentially showing that no significant improvement against (passive) supervised learning is possible (as long as no exponential number of unlabeled samples are used).
2. An upper bound in the membership query setting achieved by an efficient (poly-time) algorithm.

The achieved bounds depend on the bias $p$ (of the target halfspace $w^Tx+p\geq 0$), which make these results target/hypothesis dependent.

These results extends previous ones on the simpler problem of learning homogeneous halfspaces, where exact rates have been known and fit into similar line sof work, e.g., results under the uniform distribution on the sphere, instead of Gaussian.

**Strengths:**

* Strong near-optimal bounds for a well studied setting of actively learning halfspaces
* Devise a near-optimal poly-time algorithm in the membership query setting, while most previous work ignore computational aspects or provide only exponential time algorithms.

**Weaknesses:**

While interesting, technically deep and novel, the results are somewhat limited, due to the quite strong standard normal assumption, see question below.

Comments:
* Maybe consider changing adding something like "under the Gaussian distribution" to the title. Otherwise it might seem a bit too general.


---rebuttal---
raised score from 5 to 6.

**Questions:**

While interesting, the standard normal assumption is rather strong and limiting. Are there any possibilities to extend your results to broader families of distributions, e.g., sub-Gaussian, log-concave, ... ? Even just more general Gaussians ($\mathcal{N}(\mu,\Sigma)$)?

**Limitations:**

See question.

---

> ### Author Rebuttal · Authors · 2024-08-07
>
> We would like to thank the reviewer for the constructive feedback. We respond to the comments and questions from the reviewer as follows.
>
> We first want to make some comments on the reviewer’s summary.
>
> ## Remark on the summary:
> We do not view our work as a simple extension of previous work. Prior work had only studied the label complexity of learning *homogeneous* halfspaces. Though the label complexity of active learning homogeneous halfspaces under structured distributions (such as the Gaussian or the uniform distribution over the unit sphere) had been studied for 20 years, prior to our work there was little known for the setting of learning a general halfspace. We resolve this fundamental question in this paper, by characterizing the optimal label complexity as a function of the bias of the target halfspace. A surprising conceptual implication of our result is that, unlike learning a homogeneous halfspace, no pool-based active learning method can improve the label complexity over that of passive learning settings. Importantly, we go beyond the model of pool-based active learning and show that one can actually beat passive learning in terms of label complexity using membership queries. Specifically, we develop the first (robust to adversarial label noise) query learning algorithm with optimal query complexity. As a bonus, our algorithm is computationally efficient (not only query efficient) and potentially practical. To achieve this, we develop a number of novel technical tools that may be useful in other contexts.
>
> We next respond to the comment and weakness about the strong distributional assumption pointed out by the reviewer.
>
> ## Distributional assumptions:
>
> We start by pointing out that while we did not state the distributional assumption explicitly in the title (to avoid making the title too long),  such an assumption is clearly stated in the abstract. Since there is very little known about the label complexity of learning general halfspaces, studying the problem under the Gaussian distribution is a major first step in understanding this fundamental problem. Related to this, similar distributional assumptions were also made in early pioneering works on studying active learning homogeneous halfspaces --- such as [DKM05, BBZ07] --- and were extended by many follow-up works. At a technical level, to characterize the optimal label complexity, we develop novel techniques that we expect could be used as a foundation to study this learning problem in more general settings. With respect to our lower bound, the Gaussian distributional assumption renders the statement even stronger (since it holds even when the data is Gaussian). It has been well-known that in pool-based active learning, at least $1/p$ labels are needed to learn a $p$-biased halfspace. Despite significant interest in this question for over two decades, this lower bound has not been improved since the work [Das04]. We believe that the techniques developed in our paper to prove the lower bound could also be leveraged to establish lower bounds for other query learning problems.
>
> Regarding more general distributions: if the marginal distribution is a general Gaussian (with unknown mean and covariance), we could first use unlabeled examples to learn the marginal distribution (this can be done efficiently with polynomially many unlabeled examples); and thus reduce the problem to the case where the marginal distribution is a standard Gaussian by applying an affine transform. We believe it is also possible to design learning algorithms under all log-concave distributions, by building on our techniques. However, even in the passive learning model, efficient algorithms for agnostically learning general halfspaces are only known under the Gaussian distribution [DKS17,DKTZ22] --- with a significantly sub-optimal label complexity. We expect that there will be follow-up works that study the label complexity of learning general halfspaces under broader classes of structured distributions, such as isotropic log-concave distributions.
>
> > References
>
> >[BBZ07] Balcan, Maria-Florina, Andrei Broder, and Tong Zhang. "Margin based active learning." International Conference on Computational Learning Theory. Berlin, Heidelberg: Springer Berlin Heidelberg, 2007.
>
> >[DKM05]Dasgupta, Sanjoy, Adam Tauman Kalai, and Claire Monteleoni. "Analysis of perceptron-based active learning." International conference on computational learning theory. Berlin, Heidelberg: Springer Berlin Heidelberg, 2005.
>
>
> >[Das04]Dasgupta, Sanjoy. "Analysis of a greedy active learning strategy." Advances in neural information processing systems 17 (2004).
>
> >[DKS17]Diakonikolas, Ilias, Daniel M. Kane, and Alistair Stewart. "Learning geometric concepts with nasty noise." Proceedings of the 50th Annual ACM SIGACT Symposium on Theory of Computing. 2018.
>
> >[DKTZ22]Diakonikolas I, Kontonis V, Tzamos C, Zarifis N. Learning general halfspaces with adversarial label noise via online gradient descent. InInternational Conference on Machine Learning 2022 Jun 28 (pp. 5118-5141). PMLR.

---

> > ### Comment · Reviewer_dMkP · 2024-08-11
> >
> > Thanks for the comments. I raised my score.

---

### Author Rebuttal · Authors · 2024-08-07

We thank the reviewers for their time and effort in providing feedback. We are encouraged by the positive comments, and that all the reviewers appreciated the paper for the following (i) theoretically interesting (**dMkP,amzj,goh3,88gR,vQ1c**), (ii) technically deep and novel (**dMkP,amzj,goh3,88gR,vQ1c**), (iii) mathematically clear and rigorous (**amzj,vQ1c**). We address the individual questions and comments by the reviewers separately.

---

### Decision · Program_Chairs · 2024-09-25

**Decision:**

Accept (poster)

**Comment:**

This paper gives new positive and negative results on active learning non-homogeneous halfspaces under Gaussian distributions. The reviewers acknowledged the novelty of the lower bound (in going beyond the Dasgupta'04 lower bound), as well as the novelty of the upper bound result and algorithm under the membership query model. This can serve as a basis for follow-up works on active learning nonhomogeneous halfspaces with broader unlabeled data distributions.

We encourage the authors to incorporate the discussions in the final version, e.g. making explicit the argument of the necessity of O(min(1/p, 1/eps)) label complexity for MQ algorithms in the agnostic setting, by giving a formal proposition.